# Cell-type and subcellular compartment-specific APEX2 proximity labeling reveals activity-dependent nuclear proteome dynamics in the striatum

V. Dumrongprechachan[1,2], R. B. Salisbury[3,4], G. Soto[1], M. Kumar [1], M. L. MacDonald [3,4 ✉] & Y. Kozorovitskiy [1,2 ✉]

The vertebrate brain consists of diverse neuronal types, classified by distinct anatomy and function, along with divergent transcriptomes and proteomes. Defining the cell-type specific neuroproteomes is important for understanding the development and functional organization of neural circuits. This task remains challenging in complex tissue, due to suboptimal protein isolation techniques that often result in loss of cell-type specific information and incomplete capture of subcellular compartments. Here, we develop a genetically targeted proximity labeling approach to identify cell-type specific subcellular proteomes in the mouse brain, confirmed by imaging, electron microscopy, and mass spectrometry. We virally express subcellular-localized APEX2 to map the proteome of direct and indirect pathway spiny projection neurons in the striatum. The workflow provides sufficient depth to uncover changes in the proteome of striatal neurons following chemogenetic activation of G$\alpha_q$-coupled signaling cascades. This method enables flexible, cell-type specific quantitative profiling of subcellular proteome snapshots in the mouse brain.

[1] Department of Neurobiology, Northwestern University, Evanston, IL, USA. [2] The Chemistry of Life Processes Institute, Northwestern University, Evanston, IL, USA. [3] Department of Psychiatry, University of Pittsburgh, Pittsburgh, PA, USA. [4] Biomedical Mass Spectrometry Center, University of Pittsburgh, Pittsburgh, PA, USA. ✉email: macdonaldml@upmc.edu; Yevgenia.Kozorovitskiy@northwestern.edu

The central nervous system is made up of functionally and morphologically diverse types of neurons defined by their proteome and transcriptome[1]. Different classes of neurons throughout the brain form billions of intertwined short- and long-range synaptic connections to mediate and regulate neurotransmission, controlling sensory processing and behavior. Recent single-cell RNA sequencing (scRNA-seq) of the rodent brain revealed a vast array of neuronal types and subtypes, enabling an unbiased mapping of molecular identity and functional anatomy[2]. Although scRNA-seq provides cell-type specific information about transcription, scRNA-seq data are often uncoupled from protein translation, degradation, and subcellular localization, which influence protein function. Because proteins are the ultimate products of gene expression—the effectors that maintain and regulate cellular physiology—proteomic analyses yield a direct readout of cellular identities and states. Therefore, mapping the proteome with both cell-type and compartment specificity is crucial for understanding coordinated functions of cells and neural circuits in the vertebrate brain.

In many laboratory model systems, identifying distinct cell classes in the brain is achieved by conditional expression of site-specific recombinases and fluorescent reporters, or other effectors, under the control of a gene-specific promoter[3]. Cell isolation approaches such as laser capture microdissection (LCM)[4] or tissue dissociation followed by fluorescent cell sorting[5] are usually employed to isolate reporter-positive somata for proteomics analyses. Whole-cell patch clamp electrophysiology systems can also be used to collect cytoplasmic content for single-cell assays, supporting integrated analyses at functional and molecular levels[6]. In addition, biochemical fractionation methods are often used to access microdomain-specific proteomes, via the nuclear, membrane, or synaptosomal fractions[7]. Such preparations typically do not yield cell-type specific data. Despite improvements in protein isolation and biochemical fractionation techniques, these methods are suboptimal for mapping cell-type specific neuronal proteomes due to a substantial loss of peripheral cellular structures (e.g., dendrites and axons) that contain physiologically important information.

Recent advances in bio-orthogonal strategies for neuroproteomics facilitate the capture of proteins in cell type-specific manner. For example, genetically encoded engineered tRNA synthetase and/or tRNA pair were modified to metabolically incorporate non-canonical amino acids (NCAAs) into the proteome of a specific cell type, MetRS for methionine[8], and $SORT_{CAU}$ for tyrosine[9]. Subsequently, cell-wide labeled proteins were biotinylated via click chemistry and enriched for identification and quantification[8,9]. Because the incorporation of NCAAs depends on protein translation, this approach is not suitable for answering questions about many rapid biological processes: the experimental time window of NCAA incorporation can take up to several days[8,9].

Proximity labeling (PL) is an alternative approach to bio-orthogonal labeling. Genetically encoded labeling enzymes such as BioID[10], TurboID[11], and APEX2[12] can be expressed and localized to a specific subcellular compartment. In the presence of exogenous biotin substrates, in situ biotinylation occurs rapidly, from minutes to hours for TurboID, and within seconds for APEX2. This technique enables taking the snapshots of the proteome with restricted spatial labeling and on short temporal scales. Therefore, PL methods offer both cell-type and subcellular compartment specific information about the identified proteins. Transgenic mouse models are prevalent in neuroscience, yet, there have only been a few studies using BioID directly in the mouse brain[13–15], while HRP/APEX2-based proteomics, which offers superior labeling speed, were mostly used in primary neuronal cultures[16,17]. In the brain, changes in transcriptional and translational programs occur across short and long timescales and among multiple intermixed cell types, across their subcellular domains. Given the sophisticated organization and function of neural circuits, mapping activity-dependent changes at the proteome level and of a single cell type remains challenging. Genetically encoded fast PL is a promising solution to this problem.

In this study, we aim to characterize the proteomic landscape of the striatum. The striatum is the input nucleus to the basal ganglia, a group of subcortical nuclei responsible for motor and procedural learning, as well as goal-directed, reward-based and habitual actions[18]. Its output neurons are canonically segregated into two classes of intermingled spiny projection neurons (SPNs), called direct and indirect pathway SPNs (dSPNs and iSPNs). Both cell types are medium-sized GABAergic long-range projection neurons that compartmentalize a large subset of their excitatory synapses within dendritic spines. Dendritic architecture and basic electrophysiological properties of dSPNs and iSPNs are largely overlapping[19]. Major differences between the two SPN subtypes include the non-overlapping axonal projections, and distinct expression of dopamine (DA) and adenosine receptors. dSPNs express Drd1 DA receptors, while iSPNs express Drd2 and A2a receptors. Similar somatodendritic morphology and shared core electrophysiological properties limit the targeting methods for molecular profiling of SPNs to the expression of fluorescent reporters. RNA sequencing-based methods have been used to study the transcriptomes of SPN subtypes[2,20,21], while proteomic characterization is limited[5]. Here, we present a set of Cre-dependent APEX2 constructs for PL proteomics. We perform APEX2 PL ex vivo to map the subcellular proteomes of dSPNs and iSPNs in the mouse striatum. We show that APEX2 labeling is highly efficient, allowing enrichment and quantification of cell-type specific subcellular proteome without subject pooling. In addition, we combine the APEX workflow with a broadly used designer $G\alpha_q$-protein coupled receptor, hM3Dq, to uncover dynamic changes in the proteome of dSPNs in a short time window following chemogenetic activation.

## Results

**Cre-dependent APEX2 proximity labeling in the striatum.** APEX2 is an engineered peroxidase that can be rapidly induced to tag proteins in minutes with biotin phenol (BP) and $H_2O_2$[12]. To broadly target major subcellular proteomes in the mouse brain, we created and expressed three Cre-dependent APEX variants (Fig. 1a) that localize to the nucleus, to the cytosol, and to the membrane compartments. Targeting was attained using well-validated sorting motifs (Histone 2B fusion, H2B; nuclear exporting sequence, NES; and membrane anchor LCK sequence, respectively). In this first step, constructs were validated in Drd1-expressing dSPNs. We demonstrated Cre-dependent expression by neonatal viral transduction in the striatum of Drd1^Cre mice with adeno-associated viral vectors (AAVs) encoding one of three APEX variants and a EGFP reporter with a ribosomal skipping P2A linker (Fig. 1a). Because P2A-linked EGFP is translated separately from APEX, EGFP is expected to distribute throughout the cell, while APEX is directed to targeted subcellular domains. We verified Cre-dependent expression of our AAVs by examining the projection pattern of the EGFP reporter. Drd1^Cre+ dSPNs send axonal projections primarily to the substantia nigra pars reticulata (SNr)[22]. In all APEX variants (Fig. 1b and Supplementary Fig. 1a) EGFP expression was similar across constructs, as expected for P2A-linked EGFP, where its distribution is independent from the APEX localization and should be the same regardless of APEX targeting.

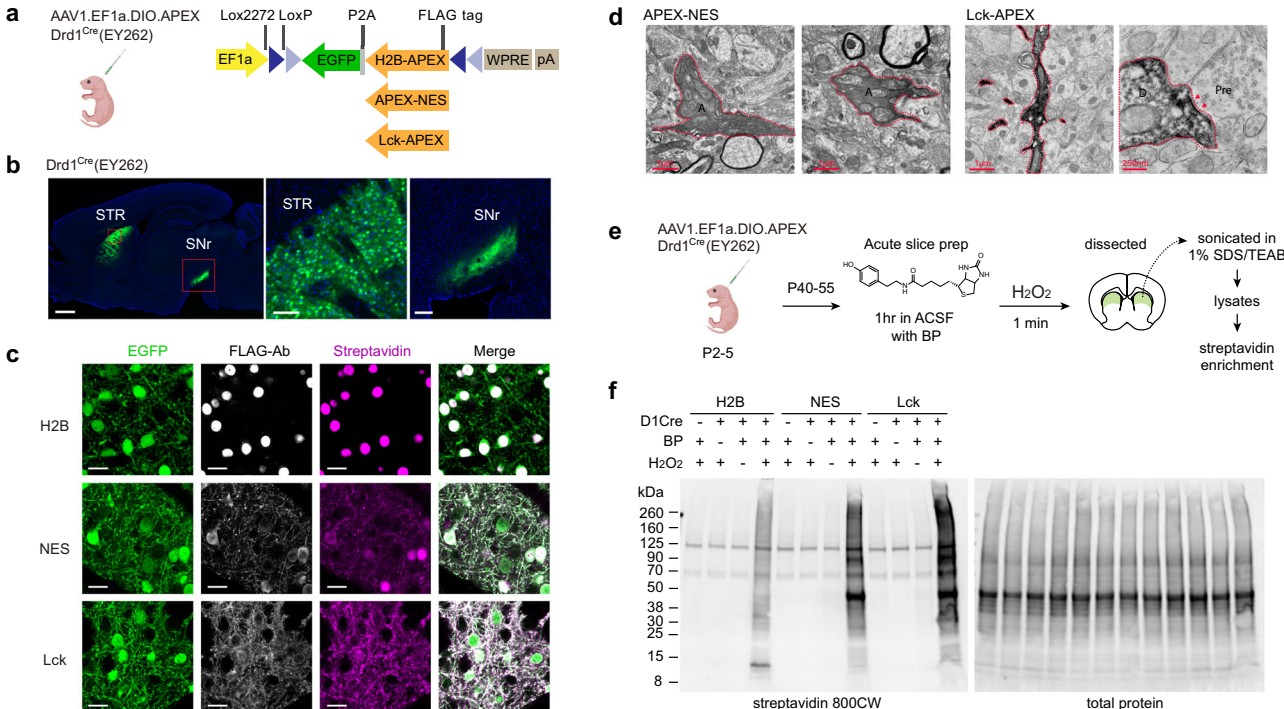

**Fig. 1 Genetically targeted subcellular protein labeling in the mouse brain. a** Design of the APEX constructs. *Left*, neonatal AAV transduction to selectively express APEX in Drd1^Cre-positive striatal neurons. *Right*, FLAG-tagged APEX constructs targeting the nucleus (H2B), the cytoplasm (NES), and the membrane (LCK). All constructs contain a P2A-linked EGFP and express under EF1a promoter. Coding sequences were inverted and flanked by lox sites in a double-floxed inverted open reading frame (DIO)-cassette for Cre-dependent expression. **b** Cre-dependent APEX-NES expression in the striatum of Drd1^Cre mouse line. *Left*, sagittal section (1 mm) of Drd1^Cre >2 weeks post AAV transduction. *Middle*, soma-fill EGFP reporter in the striatum (STR) (100 µm). *Right*, dSPN-specific axonal projections in the substantia nigra pars reticulata (SNr) (200 µm). Similar EGFP expression pattern was confirmed for all Drd1^Cre+ animals used in this study. **c** Subcellular localization of APEX constructs in striatal neurons. Confocal images of APEX-expressing neurons including EGFP, immunostained FLAG-Ab, and streptavidin for visualizing APEX subcellular localization and biotinylation patterns (20 µm). $n = 3$ animals per construct. **d** APEX labeling patterns under transmission electron microscopy (TEM). *Left*, APEX-NES containing axons (A) in the SNr show diffused labeling across the cytosol. Right, Lck-APEX containing dendrite show dense membrane-enriched signal with visible intracellular organelles. A cross section of a dendritic spine (D) with a post-synaptic density (red arrow), juxtaposed to a pre-synaptic site (Pre). APEX-labeling for TEM $n > 3$ animals per construct. $n = 2$ animals for each construct were used for TEM grid preparation. **e** Ex vivo biotinylation workflow for acute brain slices. APEX expression was achieved via neonatal AAV transduction. Some schematics were created with BioRender.com. **f** Conditional protein biotinylation in acute brain slices. APEX-mediated biotinylation requires BP and $H_2O_2$. $n = 2$ independent experiments. Source data are provided as a Source Data file.

To estimate APEX biotinylation patterns, we take advantage of retained APEX activity in paraformaldehyde (PFA)-fixed brain slices[23]. We performed protein biotinylation in PFA-fixed sections using BP. Biotinylated sections were immunostained for APEX and biotinylated proteins (Fig. 1c). Somata masks were used to compute EGFP and streptavidin signal intensity inside and outside the masks. Untargeted EGFP has similar distribution across constructs, while biotinylated protein signal differs, revealing H2B as containing the most signal inside cell bodies and LCK with most signal outside cell bodies (Supplementary Fig. 1b). In addition, a line scan across neuronal somata confirms that H2B.APEX labels proteins toward the nuclear compartment. APEX.NES broadly labels proteins in the somata, while LCK. APEX labels proteins away from the somata (Supplementary Fig. 1c). Nuclear enrichment of H2B.APEX is further confirmed by the absence of FLAG-Ab immunostained H2B signal in the SNr, in contrast to the EGFP reporter (Supplementary Fig. 1a). To further distinguish the NES and LCK constructs, we utilize APEX-peroxidase activity in fixed brain slices to selectively deposit diaminobenzidine-osmium stain and inspect localization patterns under a transmission electron microscope[12] (Fig. 1d). The NES construct showed a diffuse pattern filling axonal cross sections. In contrast, the labeling pattern for LCK in dendrites was more membrane enriched. Altogether, all APEX AAV

constructs are Cre-dependent and localize to distinct subcellular compartments and are suitable for Cre-dependent ultrastructural analyses. Importantly, APEX expression did not alter basic electrophysiological properties in neurons, compared to mCherry reporter controls (Supplementary Fig. 2).

APEX-based proteomics utilizes BP and hydrogen peroxide to label tyrosine residues via an oxidative process[24]. To ensure efficient delivery of BP in tissues, we developed an ex vivo biotinylation workflow optimized for acute brain slices (Fig. 1e). Drd1^Cre neonates were virally transduced with Cre-dependent APEX AAVs. Five to six weeks after transduction, 250 µm acute slices were prepared and incubated in carbogenated artificial cerebrospinal fluid (ACSF), supplemented with 500 µM BP for 1 h. Biotinylation was induced by transferring slices to ACSF containing 0.03% $H_2O_2$ for 1 min, followed by immersing slices in a quenching solution. EGFP-positive region in the dorsal striatum was dissected for western blot analysis (Fig. 1f). Examples of 250 µm-thick acute slices are shown in Supplementary Fig. 3a-b illustrating viral expression variability across different animals. Animals with lower EGFP expression due to suboptimal AAV injection at the time of tissue preparation were not used for subsequent experiments. We established the minimal boundary on labeling depth of APEX in acute slices using propargyl tyramide (PT), a clickable APEX substrate (Supplementary

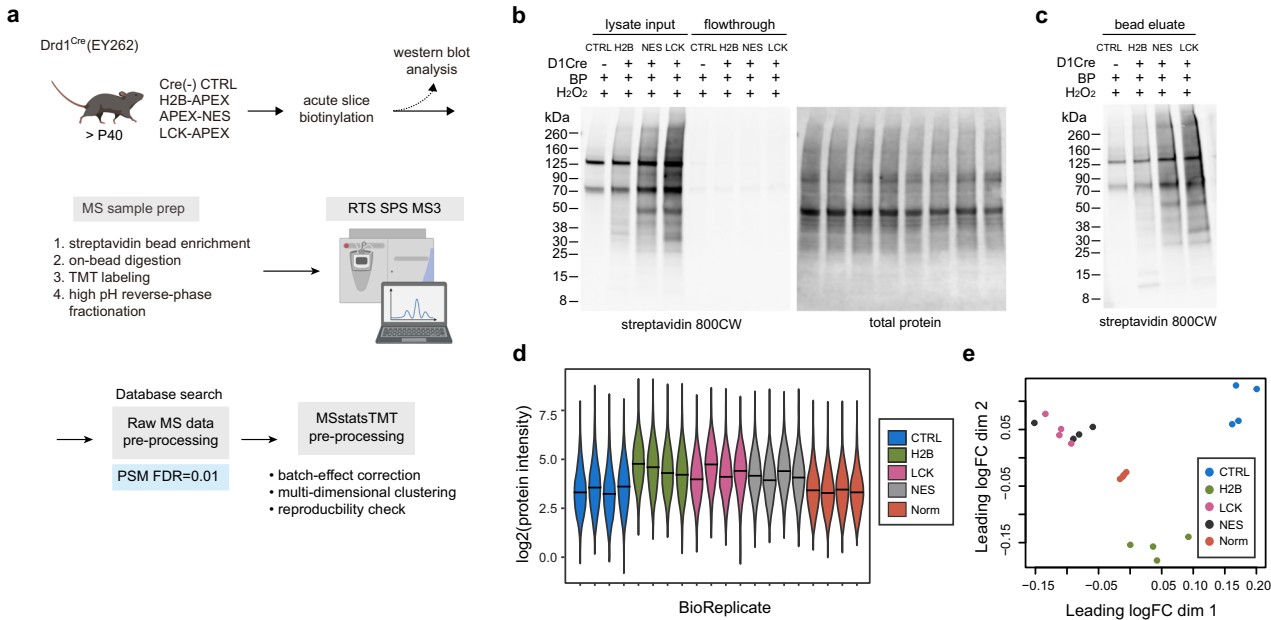

**Fig. 2 APEX2-based cell-type specific proteomics in the mouse striatum. a** Workflow for proteomics sample preparation. Following ex vivo acute slice biotinylation, biotinylated proteins were enriched with streptavidin beads. Beads were washed and digested to generated tryptic peptides. Peptides were TMT labeled and mixed equally before high pH reverse-phase fractionation. MS data were acquired using the synchronous precursor selection (SPS) MS³ method with real-time search (RTS) against the SwissProt database. Peptides were identified by Proteome Discoverer at 1% false discovery rate (FDR) and summarized by MSstatsTMT. Some schematics were created with BioRender.com. **b** Western blot analysis of biotinylated brain lysate and flow through fractions. Equal amount of proteins and streptavidin beads are used across samples in the enrichment process. Depletion of biotinylated proteins in the flow through fractions confirms successful enrichment. Enrichment process *n* = 2 independent replicates, prior to additional replications in the proteomic studies. **c** Western blot analysis of eluate fractions after streptavidin bead enrichment. Differential enrichment output reflects varying amounts of biotinylated proteins from each sample. *n* = 2 independent replicates. **d** Overall protein intensities across all samples. **e** Multidimensional scaling plot approximates expression differences between the samples for the top 500 proteins. Source data are provided as a Source Data file.

Fig. 3c-d). Acute slices were incubated in ACSF supplemented with 250 μM PT for 1 h following the same protocol as above. Labeled proteins were visualized using click chemistry with picoly-azide 594 after tissue fixation. H2B.APEX construct was used here, because nuclear localized fluorescence signal can be easily identified in thick, optically scattering tissue samples. We used a custom oblique scanning light-sheet microscope[25,26] to establish the minimum boundary on labeling depth. The deepest nuclei features were identified at ~35 μm and ~70 μm for brain slices after 1 h and 2 h click reactions, respectively. Despite going through the same protein labeling procedure (1 h PT incubation), sections that received a 2 h click reaction provided signal at a greater depth, suggesting that APEX labeling occurs at least ~70 μm from the tissue surface, most likely limited by the detection reagent. Western blot of total striatal lysates indicated that proteins were rapidly labeled within the 1 min $H_2O_2$ exposure time window. Biotinylated proteins in control samples were mostly accounted for by the endogenously biotinylated set. We also expressed and performed biotinylation in HEK293T cell culture (Supplementary Fig. 4a). Crude subcellular biochemical fractionation shows differential biotinylation by the three APEX constructs across the nuclear, cytosolic, and membrane fractions (Supplementary Fig. 4b). The pattern of H2B sample was distinguishable from that of NES and LCK—the presence and the absence of ~15 kD and ~50 kD bands, respectively—in agreement with western blots of striatal lysates and histology data.

**Mass spectrometry-based quantitative proteomics workflow for proximity labeling.** To demonstrate the application of Cre-dependent APEX AAV strategy for cell-type and subcellular compartment specific proteomics, streptavidin magnetic beads

were used to enrich biotinylated proteins from striatal lysates prepared separately for each animal, with no pooling (Fig. 2a). Western blot analysis of input, flow through, and bead eluate confirmed a successful enrichment protocol (Fig. 2b-c). Streptavidin signal in lysate input was depleted in the flow through fraction, and the presence of signal in the bead eluate verified that biotinylated proteins were properly captured. Next, we digested protein-captured beads with trypsin to generate peptides for the bottom-up proteomics approach, where peptides are analyzed by liquid-chromatography-coupled tandem mass spectrometry (LC MS/MS). To facilitate deeper proteomic coverage and quantification, peptides were labeled with TMT-10plex reagents in a randomized design (Fig. 2a, Supplementary Fig. 5a). The total of 20 samples were randomized across two 10-plex sets: four biological replicates for Cre-negative control, H2B+, NES+, and LCK+, with reference samples for each TMT 10plex set. For each TMT set, one half of the sample was desalted and run as a single injection, while the other half was analyzed after a high pH reverse-phase fractionation.

Peptide sequences were identified using the MS2 spectra, while the quantification of reporter ions was performed using the synchronous precursor selection (SPS) MS3 method[27] with the real-time search (RTS)[28]. During data acquisition, RTS triggers MS3 reporter quantification when MS2 fragmented spectra are matched to the mouse SwissProt database, improving MS3 quantification efficiency and accuracy[28]. Raw data were processed in the Proteome Discover software using the SEQUEST search engine to generate peptide spectral matches (PSMs) at 1% peptide false discovery rate (FDR). Data from fractionated samples provide a greater proteomic depth, compared to the unfractionated experiment, and are therefore used in subsequent analyses (Supplementary Fig. 5f-g). We identified a total of 4365 proteins

across both TMT 10-plex experiments (Supplementary Data 1). We used the MSstatsTMT R package to perform protein-level summarization and quantification[29]. A total of 2191 proteins were quantified (Supplementary Data 1). The overall log2 protein intensities for each biological replicate were plotted in Fig. 2d. TMT batch effects were corrected protein-by-protein based on protein signal in the reference channel. A multidimensional scaling (MDS) plot after reference-channel normalization revealed that APEX construct replicates cluster together (Fig. 2e). This normalization is necessary in TMT experiments: an MDS plot of un-normalized data showed separation between two TMT batches (Supplementary Fig. 5b). To evaluate the reproducibility of this workflow, we plotted an example of a multi-scatter plot of H2B biological replicates. Pearson correlation of log2 protein intensities were highly correlated ($r > 0.9$) (Supplementary Fig. 5c). The distribution of coefficient of variation percentages (%CV) for each condition were similar, with a mean(%CV) of <10% (Supplementary Fig. 5d-e). Collectively, biotinylation, streptavidin enrichment, and MS sample preparation procedure were highly reproducible.

**Genetically targeted proximity labeling approaches for cell-type specificity and subcellular localization.** To filter out proteins that were enriched non-specifically by the beads, we defined the first filter based on comparisons between Cre-positive samples and Cre-negative control samples using a pairwise moderated $t$-test implemented in MSstatsTMT (Fig. 3a). Proteins that were not statistically significantly enriched in the positive direction were discarded as nonspecific binders (Fig. 3b, Supplementary Fig. 6a). Gene names will be used to refer to gene products for ease of reading. Some examples included known endogenously biotinylated proteins (*Acaca, Pccb, Pcca, Pc, Mccc1*). Retained proteins with log2-fold change > 0 and adjusted $p$ values or $q$ values < 0.05 were considered as cell-type specific and enriched due to the biotinylation process (Supplementary Data 2). In an agreement with western blot and histology, the majority of the proteins detected in Cre-positive samples were retained as cell-type specific (Fig. 3b). Next, we aimed to define a second, stringent cutoff for the nuclear and membrane-enriched proteome using $t$-tests (Fig. 3c). We performed pairwise comparisons for H2B-NES and H2B-LCK to generate nucleus-enriched and membrane-enriched protein lists, with multiple comparison adjustment (Supplementary Data 2). We obtained 219 nucleus-enriched proteins and 63 membrane-enriched proteins after applying the second filter. The choice of a reference compartment was important for the statistical comparison. Consistent with its broad cellular distribution, LCK-NES comparison did not yield any statistically significant enrichment for either compartment after FDR adjustments ($q$ value > 0.05).

The lower statistical power for this comparison is reflected by a modest separation between the LCK-NES log2 fold changes. Because many proteins are known to localize to both the cytosol and the membrane, the NES and LCK constructs are capable of labeling a similar set of proteins (Supplementary Fig. 6b). A rank plot between NES and LCK indicated a relative enrichment of membrane proteins toward the LCK construct. This suggests that the two constructs can be used to compare the membrane and cytosolic compartments, but more replicates are likely needed to increase the statistical power for smaller protein log2FC. Alternatively, a rank-based or static cutoff can be used instead[30] (Supplementary Fig. 6c-d).

To examine whether H2B-enriched and LCK-enriched proteins are overrepresented in any particular subcellular compartment, we used gene ontology analysis with the total identified proteins as background (Supplementary Data 3). Indeed, the top terms with lowest FDR for each protein list were nucleus- and membrane-related terms, confirming that the two APEX constructs labeled proteins in different compartments (Fig. 3d). Next, we created log2 fold change vs rank plots with nucleus or membrane annotations. As expected, proteins with greater fold change are more likely to have prior UNIPROT nucleus or membrane annotations for H2B and LCK, respectively (Fig. 3e, f). Consistent with recent proteomic profiling of Drd1[Cre+] striatal nuclear proteome using fluorescent nuclei sorting, we found that Drd1[Cre+] nuclear proteome highly expressed *Hnrnpa2b1* and *Hnrnpd* protein network[5]. Additional experiments are needed to determine whether identified nuclear proteins were generally widely expressed nuclear proteins or exclusively expressed by dSPNs. In the LCK rank plot, several proteins do not list plasma membrane as their primary cellular component annotation in the UNIPROT database, including *Synpo, Dock3, and Actn4*. *Synpo* is an actin associated proteins found in post-synaptic densities and dendritic spines, implicated in long-term spine stability[31]. *Dock3* is a member of guanine nucleotide exchange factors that regulates membrane-associated protein, *Rac1*. It was also shown to interact with the NR2B subunit of glutamatergic NMDA receptors[32]. *Actn4* is a filamentous actin-binding protein containing a group1-mGluR binding domain[33]. *Actn4*-mGluR interactions have been implicated in the remodeling of dendritic spine morphology[33]. Although *Dock3* and *Actn4* distribute throughout the cytoplasm, both were known to interact with membrane proteins, suggesting a potential future application of this workflow for in situ protein-protein interaction studies. Altogether, we have established an APEX-based approach using LCK/NES and H2B.APEX to map the somatodendritic and the nuclear proteome with cell-type specificity in the mouse brain, with no subject pooling.

**Cytosolic and nuclear proteome of striatal direct and indirect pathway neurons.** Our next goal is to compare the baseline proteome of striatal direct and indirect pathway spiny projection neurons (dSPNs and iSPNs). SPNs represent a strong model for further validation of cell-type specific proteomics approach for several reasons. First, dSPNs and iSPNs are broadly similar, intermixed, highly abundant, and approximately equal in numbers, together comprising ~95% of the striatum. Therefore, no additional changes need to be made in the sample preparation process to compare the two cell types. Second, molecular differences between the two SPN subtypes are known at the transcriptomic level, and data are readily accessible for benchmarking. Third, the proteome of SPNs is not yet extensively characterized.

To map both the cytosolic and the nuclear proteome of SPNs, we express APEX.NES and H2B.APEX in striatal Drd1[Cre] and A2a[Cre] neurons via neonatal viral transduction to target dSPNs and iSPNs, respectively (Fig. 4a). To ensure that we can properly target iSPNs as we did for dSPNs, we evaluated the anatomical expression pattern of APEX and EGFP reporters in A2a[Cre] tissues. Unlike dSPNs (Fig. 1b), A2a[Cre+] iSPNs primarily innervate the external globus pallidus (GPe) as expected for this cell type (Fig. 4c). In addition, no qualitative differences were observed in the APEX biotinylation pattern between Drd1[Cre+] dSPNs and A2a[Cre+] iSPNs (Fig. 4b, d, Supplementary Fig. 7a), confirming that we can use the AAV APEX strategy in both SPN types. Following the same workflow for sample preparation (Fig. 4e), we used a TMTPro 16plex study design ($n = 3$ for APEX.NES and $n = 5$ for H2B.APEX per each SPN subtype) (Supplementary Fig. 7d) and high pH reverse-phase fractionation before MS analysis to generate this dataset. For reproducibility evaluation, we ensured that all samples are appropriately biotinylated (Supplementary Fig. 7b). Peptide quantification assay was also used to assess the relative enrichment yield after on-bead digestion (Supplementary

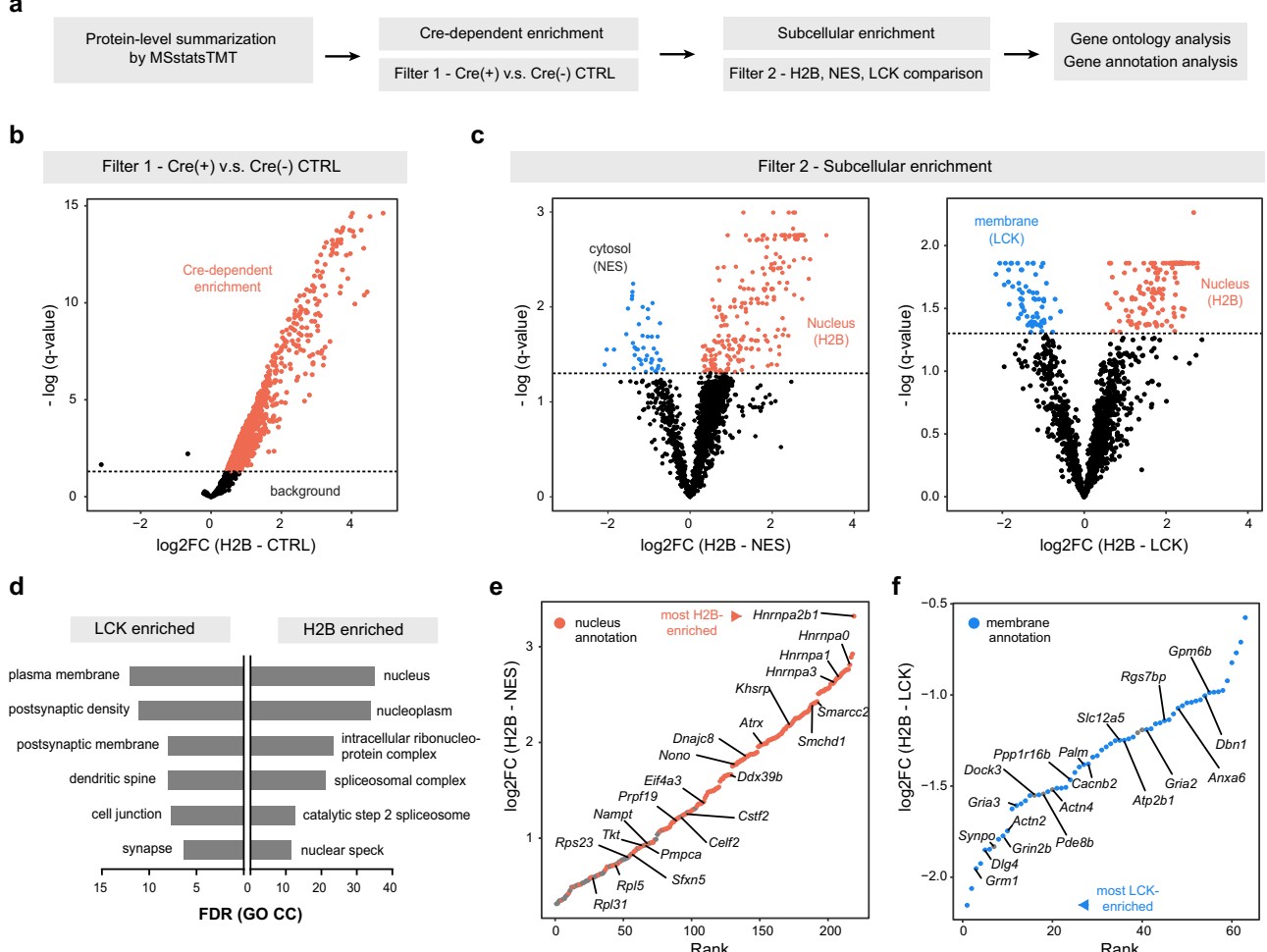

**Fig. 3 In situ proximity labeling generates cell-type and subcellular compartment specific proteomes. a** Statistical analysis workflow. Moderated $t$-test for significance with Benjamini–Hochberg (BH) multiple hypothesis correction is used as a cut-off. Filter 1 is the first comparison to determine the degree of protein enrichment above nonspecific background. Cre-positive samples are compared against Cre-negative control to remove enrichment contaminants. Filter 2 is used with proteins that were retained after the first filter to determine the extent of subcellular enrichment. **b** Differential enrichment analysis for Drd1$^{Cre}$ direct-pathway spiny projection neurons (dSPN) (H2B – CTRL). Proteins significantly enriched above background are indicated by red dots ($q$ value < 0.05 and log2FC > 0). **c** Differential enrichment analysis for Drd1$^{Cre}$ dSPN (H2B – NES) and (H2B – LCK). *Left*, nucleus enriched proteins (H2B) are indicated by red dots. *Right*, nucleus enriched proteins (H2B), red dots; membrane-enriched proteins (LCK), blue dots. **d** Gene ontology cellular component (GO CC) analysis for H2B-enriched and LCK-enriched proteins. **e** log2FC vs rank plot for H2B-enriched nuclear proteins. Proteins with UNIPROT nucleus annotation, red dots, a subset of proteins labeled. Positive log2 fold change (H2B-NES) indicates biased enrichment toward the nucleus. **f** log2FC vs rank plot for LCK-enriched membrane proteins. Proteins with UNIPROT membrane annotation, blue dots, a subset of proteins labeled. Negative log2 fold change (H2B-LCK) indicates biased enrichment toward the membrane. Source data are provided as a Source Data file.

Fig. 7c). All Cre+ samples yield greater amount of peptide signal compared to Cre-negative control.

To make a quantitative comparison across compartments and cell types, protein summarization was carried out in MSstatsTMT with peptide-level global normalization. We also performed protein-level median normalization to align median of protein abundance across replicates and conditions (Supplementary Fig. 8a). We identified 5668 proteins and quantified 2332 proteins (Supplementary Data 4). We filtered out 239 non-specific enrichment contaminants using a list generated based on our prior experiments. We performed moderated $t$-tests between NES and H2B compartments (i.e., Drd1$^{Cre}$.H2B – Drd1$^{Cre}$.NES, and A2a$^{Cre}$.H2B – A2a$^{Cre}$.NES) to assign proteins into two groups, nucleus/H2B and cytosol/non-H2B enriched. Proteins with $q$ value < 0.05 and with log2FC (H2B – NES)>0 were marked as nucleus-enriched proteins (Supplementary Fig. 8d). Hierarchical clustering heatmaps of the NES and H2B show similarity across

biological replicates and conditions (Supplementary Fig. 8b,c). Gene ontology analysis of 458 H2B-enriched proteins primarily shows nuclear-related GO terms, demonstrating that our subcellular compartment cutoff analysis works well (Fig. 4f, Supplementary Data 6).

To test whether proteins are differentially expressed between dSPNs and iSPNs, we used moderated $t$-tests to compare the H2B-enriched nuclear proteins (A2a$^{Cre}$.H2B – Drd1$^{Cre}$.H2B) and the non-H2B cytosol enriched proteins (A2a$^{Cre}$.NES – Drd1$^{Cre}$.NES) (Fig. 4g, h, Supplementary Data 5). A modest log2FC among proteins between dSPNs and iSPNs were observed in this dataset. Therefore, we consider proteins differentially expressed with $p$ value < 0.05. Drd1 and A2a receptors were correctly identified in the cytosolic proteome of dSPNs and iSPNs, respectively (highlighted in Fig. 4g). As for the Drd1$^{Cre+}$ nuclear proteome detected in the prior dataset (Figs. 2–3), the majority of highly abundant nuclear proteins were general nuclear markers

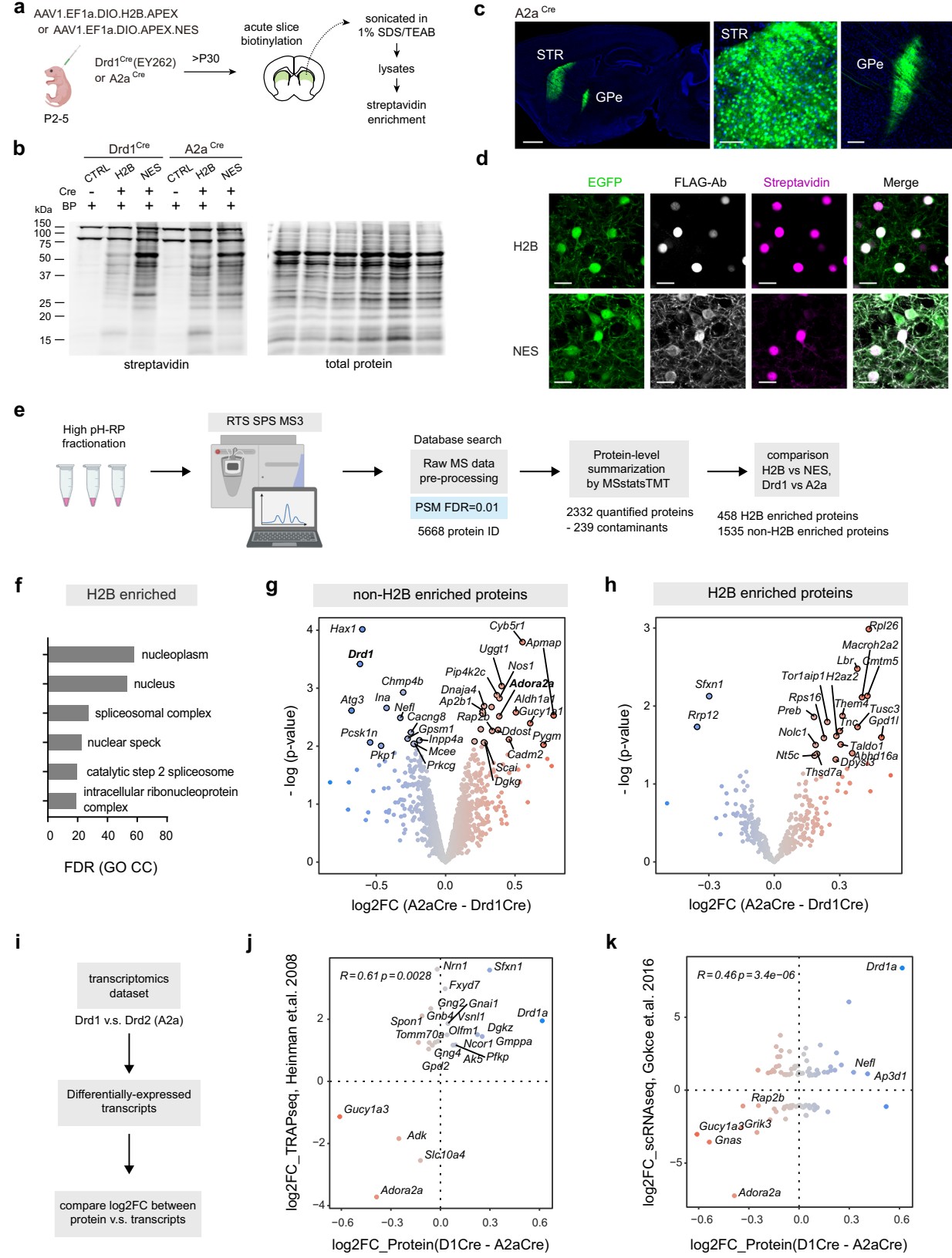

for both SPNs, measured by A2a$^{Cre}$ – Drd1$^{Cre}$ comparison (e.g., *Hnrnpa2b1*, log2FC = −0.043, $p = 0.607$, and *Hnrnpd*, log2FC = −0.011, $p = 0.89$). Apart from the two canonical markers for SPNs, we investigated whether there are other known differentially regulated proteins detected in the dataset. To accomplish this, we compared the proteomic data with public transcriptomic

data for SPNs (Fig. 4i-k). We restricted this comparison to a previously reported set of highly differentially expressed transcripts. We compared log2FC protein to transcripts using data generated by two techniques including translating ribosome affinity purification (TRAPseq) and scRNAseq[20,21]. A moderate correlation was observed among proteins and transcripts across

**Fig. 4 APEX-based comparison between the nuclear and cytosolic proteomes of dSPNs and iSPNs. a** Targeted expression of APEX in direct and indirect-pathway spiny projection neurons (dSPNs and iSPNs). Some schematics were created with BioRender.com. **b** Western blot analysis of biotinylated striatal lysates prepared from Drd1[Cre] and A2a[Cre] mouse lines. Western blot analysis of these samples was performed once prior to orthogonal confirmation by proteomics. **c** Cre-dependent APEX-NES expression in the striatum of A2a[Cre] mouse line. *Left*, sagittal section (1 mm) of A2a[Cre] > 2 weeks post AAV transduction. *Middle*, soma-fill EGFP reporter in the striatum (STR) (100 μm). *Right*, iSPN-specific axonal projections in the external globus pallidus (GPe) (200 μm). $n = 2$ animals per construct. Similar EGFP expression pattern was confirmed for all A2a[Cre+] animals used in the proteomic experiment. $n = 3$ and 5 for NES and H2B, respectively. **d** Subcellular localization of APEX constructs in striatal iSPNs. Confocal images of APEX-expressing neurons including EGFP, immunostained FLAG-Ab, and streptavidin for visualizing APEX subcellular localization and biotinylation patterns (20 μm). $n = 2$ animals per construct. **e** Statistical analysis workflow for comparing dSPN and iSPN proteomes. Some schematics were created with BioRender.com. **f** Gene ontology analysis of the 458 H2B-enriched nuclear proteome generated from H2B – NES comparison. Non-H2B enriched proteins are considered as a part of the cytosolic proteome. **g** Differential expression analysis for the non-H2B enriched cytosolic proteome (A2a[Cre].NES – Drd1[Cre].NES). **h** Differential expression analysis for the H2B-enriched nuclear proteome (A2a[Cre].H2B – Drd1[Cre].H2B)**. i** Workflow for correlation analysis between proteins and transcripts. **j** Correlation between log2FC (dSPN – iSPN) for proteins and translating mRNA TRAPseq dataset. **k** Same as (**j**) for mRNA expression from single-cell RNAseq dataset. Source data are provided as a Source Data file.

the two datasets (Pearson's $r = 0.61$ $p = 0.0028$, $r = 0.46$ $p < 0.001$, for TRAPseq and scRNAseq, respectively. Supplementary Data 10). In addition to DA and adenosine receptors, we found several genes that are differentially expressed at both transcript and protein levels, including *Sfxn1*, *Nefl*, *Dgkz*, *Gmppa* for dSPNs, and *Gucy1a3*, *Grik3*, *Rap2b*, *Adk* for iSPNs. Evidence of differentially expressed genes at the protein level implies relative functional significance for dSPN and iSPN physiology. Given a growing catalog of scRNAseq data for many cell types in the brain, cell-type specific proteogenomic analyses—yielding the correlation between the proteome and the transcriptome for a cell type—are useful, and especially so for cell types with limited proteomic annotation.

**Activity-dependent nuclear proteome dynamics in striatal dSPNs**. Molecular adaptations are required for long-lasting changes in synaptic plasticity and learning. The molecular basis of plasticity involves cell-type specific modifications of transcriptome and proteome on multiple timescales, from minutes and hours to days[34]. We showed that APEX biotinylation can efficiently capture proteomic snapshots in a short time window in the acute brain slice preparation. Next, we aim to test whether our ex vivo labeling strategy provides sufficient depth and precision to reveal activity-dependent changes in the proteome. As a proof of principle, we chose to examine early changes in the nuclear proteome of dSPNs, after a chemogenetic manipulation using an engineered Designer Receptors Exclusively Activated by Designer Drugs (DREADD), hM3D, a Gα$_q$-coupled receptor broadly used in neuroscience studies[35]. For the proteomic snapshot, we selected the time point of 2.5–3 h following an in vivo neural circuit manipulation, because immediate early genes (IEGs) are expected to be upregulated in the nucleus. Chemogenetic activation of hM3D-expressing neurons by its synthetic ligand clozapine N-oxide (CNO) is cell-type selective, allowing a direct association between cell type specific changes in the proteome, neuronal activity, and, in many cases, animal behavior. We co-expressed hM3Dq, along with H2B.APEX in striatal Drd1[Cre] neurons via neonatal viral transduction (Fig. 5a). hM3Dq has been extensively characterized in dSPNs[36]. Activation of hM3Dq by CNO increases dSPN excitability by raising intracellular calcium, in turn, enhancing their firing rate ex vivo[36]. To validate our experimental paradigm, we first evaluated AAV co-transduction efficiency in Drd1[Cre+] dSPNs. The co-expression of hM3Dq- or fluorophore control and H2B.APEX was nearly 100%, and it did not affect H2B.APEX biotinylation activity or localization (Fig. 5b, Supplementary Fig. 9a, b). CNO administration resulted in an upregulation of the immediate early gene, cFos (*Fos*) (Supplementary Fig. 9c, d), as well as increased locomotor activity evaluated by the open-field locomotion assay (Supplementary

Fig. 9e-g). Our results indicate that hM3Dq-APEX co-expression is attainable in dSPNs and can be used to assay proteomic changes in response to activation by CNO.

To investigate the proteome following chemogenetic activation of dSPNs, we used a TMT-11plex study design (2 TMT sets, $n = 2$ reference channels, $n = 7$ hM3Dq.H2B, $n = 7$ mCherry.H2B, $n = 3$ APEX.NES, $n = 3$ Cre-negative control) (Supplementary Fig. 10a). We included Drd1[Cre+] APEX.NES samples in this experiment as a compartment reference for H2B.APEX, so that we can assign subcellular annotation (i.e., nucleus or cytosol enriched). All protein and peptide samples were analyzed using western blotting and peptide quantification assay prior to MS analyses, checking for sample reproducibility (Supplementary Fig. 10b, c). Similar to previous experiments, protein summarization was performed by MSstatsTMT with both peptide- and protein-level median normalization (Fig. 5c, Supplementary Fig. 11c, Supplementary Data 7). A total of 3934 proteins were quantified across two TMT sets. Following the same filtering procedure, hM3Dq, mCherry, or NES were compared against Cre-negative control (Supplementary Fig. 11a). 3670 proteins were retained as cell-type specific. Retained proteins in hM3Dq or mCherry samples were compared against NES samples to generate H2B enriched nuclear proteome (Supplementary Fig. 11b). We identified 617 H2B enriched nuclear proteins and 3053 cytosolic proteins. Hierarchical clustering heatmap shows similarity across proteins, biological replicates, and conditions (Fig. 5d). Gene ontology analysis of H2B enriched proteins confirms successful classification of nuclear proteins (Fig. 5e, Supplementary Data 9).

To identify differentially regulated proteins in dSPNs after hM3Dq-mediated modulation, we used moderated t-tests (hM3Dq – mCherry) and considered proteins to be differentially regulated when $q$ value < 0.05. As expected, the nuclear proteome of hM3Dq and fluorophore control samples were well correlated (Fig. 5f), with differentially regulated proteins identified across high and low abundant proteins. A total of 127 nuclear and 64 cytosolic proteins were upregulated, while 80 nuclear and 129 cytosolic proteins were downregulated (Fig. 5g, h, Supplementary Data 8). M3 muscarinic receptor, *Chrm3*, presents the largest fold change between hM3Dq and mCherry samples. This is most likely due to the overexpression of hM3Dq, which is a modified form of the human M3 muscarinic receptor[37] with a high sequence similarity to the mouse receptor, serving as an internal control for our data. We also expected an upregulation of multiple IEGs, which consist of many transcription factors, in this time period after dSPN activation. Notably, known IEGs such as *Junb*, *Egr4*, *Nr4a1*, and *Arc* were upregulated in our dataset. Because *Junb* has the largest fold change, we repeated this chemogenetic experiment and performed immunofluorescence

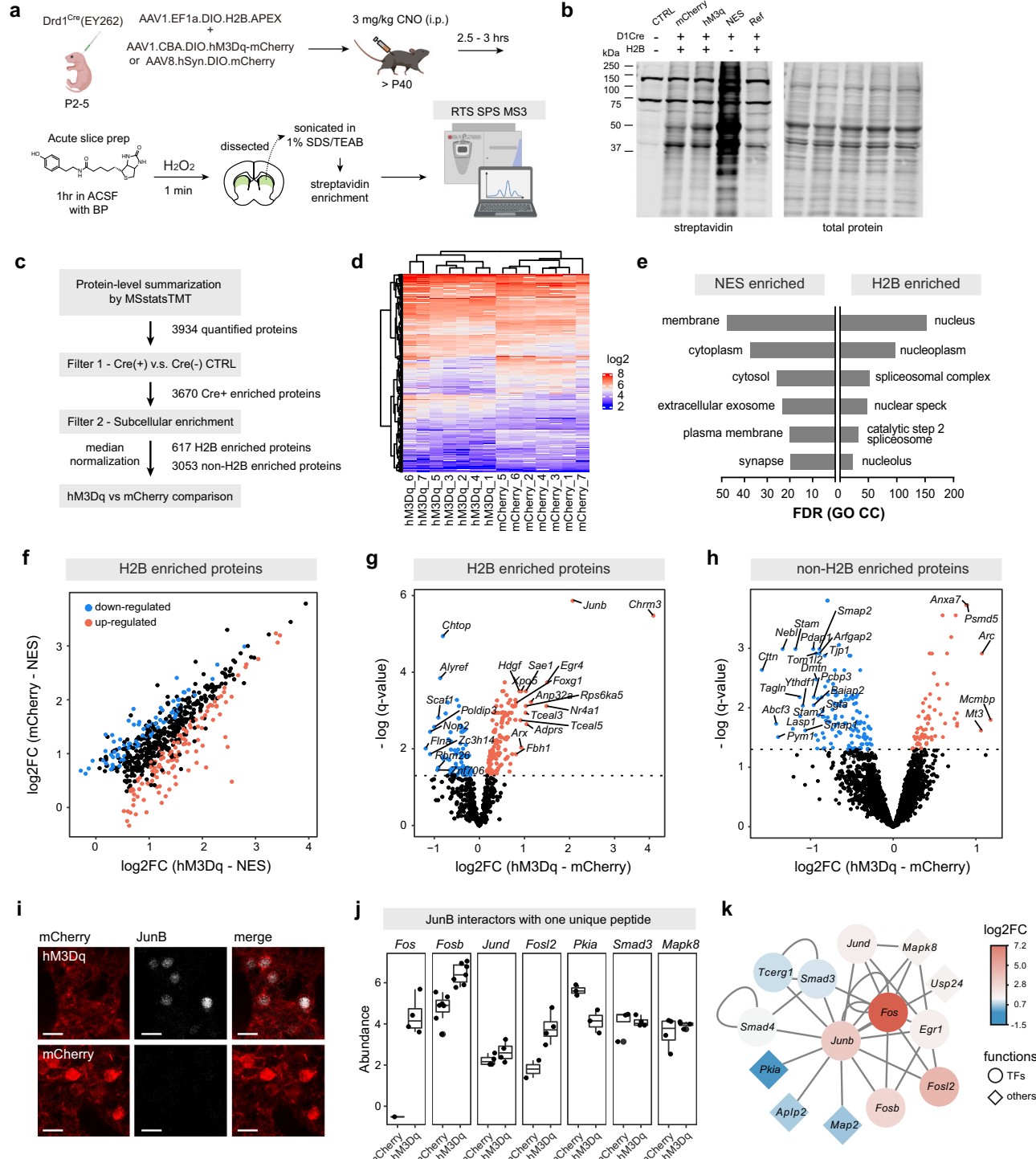

staining for further validation. We confirmed that *Junb* is upregulated in hM3Dq-expressing dSPNs, consistent with the proteomic data (Fig. 5i).

In the MSstatsTMT workflow, we opted to only summarize proteins with at least two unique features (i.e., PSMs), a typical standard used in the proteomics field. Consequently, many proteins that were reliably quantified with only one unique PSM were omitted in our prior statistical analysis. We decided to slightly relax this criterion and examined proteins that were quantified with one unique PSM of high confidence, passing 1% FDR cutoff. We found additional IEGs along with other *Junb* interactors, including *Fos* (cFos), *Fosb*, *Jund*, *Fosl2*, *Pkia*, *Smad3*,

and *Mapk8* (Fig. 5j). We plotted a *Junb* protein interaction network in Fig. 5k to better visualize functional annotations along with the corresponding log2FC. Other than transcription factors, *Pkia* was downregulated in the nucleus of dSPNs after hM3Dq activation (two-sided *t*-test, $p = 0.0136$). *Pkia* is a potent inhibitor of the nuclear protein kinase A (PKA) activity and is believed to facilitate the export of PKA from the nucleus to the cytoplasm by recruiting nuclear export machinery[38]. To what extent PKA signaling pathway is involved in the $G\alpha_q$ signaling-mediated activation of dSPNs remains to be elucidated in future studies; yet $G\alpha_q$ signaling can modulate PKA activity via muscarinic acetylcholine receptors in hippocampus[39]. Altogether, we

**Fig. 5 Differential expression analysis of the dSPN proteome following CNO-induced activation of hM3Dq. a** Experimental paradigm. Targeted co-expression of APEX and hM3Dq or fluorophore control in Drd1[Cre+] dSPNs. Clozapine N-oxide (CNO) was administered (3 mg/kg i.p.) to selectively activate dSPNs via Gα$_q$-coupled signaling cascades. Following CNO treatment, acute slice preparation and ex vivo biotinylation were performed to measure proteomic changes at 2.5–3 h post CNO administration. Some schematics were created with BioRender.com. **b** Western blot analysis of biotinylated striatal lysates for samples used in this dataset. Western blot analysis of these samples was performed once prior to orthogonal confirmation by proteomics. **c** Statistical analysis workflow for comparing the effect of CNO-induced changes in the dSPN proteome between hM3Dq-expressing and fluorophore control samples. **d** Hierarchical clustering heatmap of proteins and biological replicates. **e** Gene ontology analysis of H2B-enriched and NES-enriched protein lists. **f** Scatter plot of log2FC (mCherry-NES) vs log2FC (hM3Dq-NES) for the H2B-enriched nuclear proteome. Blue and red dots are downregulated and upregulated proteins, respectively (q value < 0.05). **g** Differential expression analysis for the nuclear proteome (hM3Dq – mCherry) (dotted line: q value = 0.05). **h** Differential expression analysis for the cytosolic proteome (hM3Dq – mCherry) (dotted line: q value = 0.05). **i** Immunostaining of *JunB* in the striatum 2.5 h post CNO administration (20 μm). $n = 3$ animals from two independent experiments. **j** Reporter ion quantification of JunB interacting proteins that were identified only by one unique peptide. For each protein, $n$ reflects the number of samples where this protein was quantified, for mCherry and hM3Dq conditions, respectively: *Fos* $n = 1, 4$; *Fosb* $n = 7, 7$; *Jund* $n = 4, 4$; *Fosl2* $n = 2, 4$; *Pkia* $n = 3, 3$; *Smad3* $n = 4, 4$; and *Mapk8* $n = 4, 4$. The upper and lower bounds of the box represent 75th and 25th percentiles, whiskers extend from minimum to maximum. The horizontal center line is the median. **k** Cytoscape network of JunB protein interactions detected in the dataset (color: log2FC (hM3Dq – mCherry)). Source data are provided as a Source Data file.

demonstrated that the APEX2-based PL can be used to interrogate activity-dependent changes in the neural proteome with cell type and subcellular compartment specificity.

## Discussion

To date, only a few techniques are available to map cell-type and microdomain-specific neuroproteomes in the mouse brain. In this study, we demonstrate that the APEX2 PL workflow is highly efficient for profiling subcellular proteome of neurons across the nucleus, the cytoplasm, and the intracellular membrane compartments. We map the compartmental proteome of two similar neuron types—dSPNs and iSPNs in the dorsal striatum—without the need to perform biochemical or organelle isolation. AAV transduction of APEX2 generates sufficient peptide output, without subject pooling prior to the streptavidin enrichment and TMT labeling. Together with peptide fractionation, over 3000 proteins can be quantified with the TMT multiplexing approach. Multiple technical replicates and exclusion lists can be used to increase proteome coverage. Pooling across animals may be necessary for smaller brain regions or for very low abundance subcellular compartments. The amount of beads can also be systematically adjusted to improve yield, because competition from endogenously biotinylated proteins could affect the ranking of proteins in the data. This could be particularly valuable for samples where correlative information is difficult to obtain from e.g., transcriptomics datasets. Here, we did not vary bead concentration, because we were able to identify many known positives for a given APEX construct, cell class, and activity states using our statistical analysis approach. Complementary to our work, a recent preprint reports the use of the cytosolic APEX to map the proteome of midbrain dopaminergic neurons in mice[40]. In addition to APEX, BioID/TurboID is another type of PL enzyme. Despite having a greater labeling time window, BioID/TurboID can label proteome in vivo with high spatial precision[13–15]. Apart from PL approaches, non-canonical amino acid tagging has been used to profile broad cell classes (e.g., the hippocampal excitatory neurons vs cerebellar inhibitory neurons[8], and striatal GABAergic neurons[9]). For organelle-specific proteome, cell-type specific GFP-tagged mitochondria can be purified from MitoTag mice, with a recent study showing distinction of the mitochondrial proteome among cerebellar Purkinje cells, granule cells, and astrocytes[41]. With a growing toolbox of genetically encoded proteomic reporters, our workflow is one of the most sensitive approaches for affinity enrichment-based and cell-type specific proteomics in the mouse brain.

Validation of cell-type specific proteomic data remains challenging. Our approach enables quantification of relative protein abundance across cell types, while orthogonal targeted techniques are still required to determine whether identified proteins are exclusively expressed by only that cell type. Our dataset serves as one of the few cell-type specific proteomic maps of dSPNs, and iSPNs with the proteome coverage spanning the somatodendritic and nuclear compartments. We make use of existing single-cell[21] and single cell-class[20,42] specific transcriptomic data to support our findings. We found a group of proteins that are differentially expressed across two types of SPNs—*Sfxn1, Nefl, Dgkz, Gmppa* for dSPNs, and *Gucy1a3, Grik3, Rap2b, Adk* for iSPNs. For example, *Adk*, adenosine kinase, converts adenosine and ATP to ADP and AMP, facilitating the clearance of extracellular and intracellular adenosine[43]. In the striatum, DA and adenosine bidirectionally regulate activity and synaptic plasticity of SPNs[44,45]. Specifically, adenosine enhances iSPN plasticity via A2a receptors, while DA decreases plasticity via Drd2 receptors. It is thought that A2a receptors are activated by adenosine generated by iSPN-specific and synaptically localized ecto-5′-nucleotidase *Nt5e*[46], while adenosine tone is cleared by astrocyte-derived *Adk*[47]. In our study, the relative abundance of *Adk* in iSPNs points toward an intrinsic self-regulation of adenosine signaling, in addition to glial control. Although our study provides protein-level evidence for many differentially expressed transcripts, the roles of numerous proteins in regulating SPN physiology are not fully understood and remain to be elucidated. The current datasets provide numerous targets for directed follow up investigations. Overall, this Cre-dependent AAV transduction workflow can be applied to many available Cre-driver mouse lines to establish and compare the proteome of various cell types.

Changes in the proteome and transcriptome occur rapidly across neuronal compartments in response to plasticity-promoting factors. This coupling between neuronal activity and gene expression varies among cell classes and at single-cell level. While transcriptomic and chromatin-state analyses of activated neurons reveal primary and secondary waves of activity-dependent gene expression[48], levels of protein products are often inferred. Here, we demonstrate a powerful methodology to directly measure activity-dependent changes in the proteome with cell-type and compartment specificity. We leverage the subcellular localization and speed of the APEX-based strategy to capture a snapshot of the early changes in the nuclear proteome of dSPNs after chemogenetic activation with hM3Dq[35,49,50]. We measure protein products of multiple IEGs a few hours after neuronal activation in vivo. These include the AP1, *Egr* transcription factor family and *Arc* transcriptional regulator. IEGs are the primary response genes that are thought to control the secondary response genes to elicit long lasting functional and

structural changes in neurons[48]. The extent to which various IEGs are utilized across different neuronal types, as well as their downstream gene targets, remain to be explored. While the development of DREADDs has transformed behavioral neuroscience research, we have limited insight on how gene expression programs are shaped by DREADD activation. Time course studies involving proteomic, transcriptomic, and chromatin-state readouts are likely needed to establish the complete picture of activity-dependent gene expression dynamics and regulation in response to specific stimuli.

The complications of using the APEX approach to map activity-dependent proteome include the delivery of BP and the toxicity of $H_2O_2$ exposure. These constraints limit the fully in vivo APEX-based workflow[51], but our study demonstrates an efficient approach where in vivo proteomic changes are captured in an ex vivo preparation, leveraging the labeling speed of APEX. Thus, the strategy can support activity-dependent investigations of proteomic changes. Unlike transcriptomics or other proteomics techniques, our APEX workflow directly provides evidence of relative protein abundance within both a restricted spatial domain as well as a short temporal window. While the current work demonstrates activity-dependent cell-type specific profiling for the nuclear proteome, the method is modular and applicable to other cellular compartments. Although we did not isolate SPN terminals from GPe or SNr in this study, future work investigating the proteomes of SPN axons may uncover differences that relate to segregated projection targeting, developmental dynamics, or changes in disease models. If multiple APEX constructs (e.g., H2B, NES, LCK) are being assayed in parallel for a case-control experiment, it is plausible that protein translocation events can be detected, similar to previously published cell culture experiments[52,53]. The APEX-based strategy is complementary to the non-canonical amino acid tagging approach (NCAA). While APEX takes the snapshot of the proteome, NCAA methods directly assay translational dynamics of the newly synthesized proteome.

In summary, APEX-based proteomics is a strong candidate for mapping the proteome of cell-type specific organelles, including compartments that cannot be easily isolated with conventional biochemical fractionation methods[17,30,54]. The dramatically reduced sample amount requirements of the APEX2+TMT labeling workflow opens the possibility for tandem affinity enrichment strategies to profile both compositional and post-translational proteome of specific neural circuits. The expansion in the availability of Cre driver lines for rodents and growing genetic traction over previously inaccessible animal models, along with the increase in the number of APEX toolkits[3,23] and the validated workflow presented here, positions APEX-based proteomics to become the major cell-type specific proteomics approach for diverse neuroscience applications.

## Methods

**Plasmid construction and AAV preparation**. APEX2-NES-P2A-EGFP, LCK-APEX2-P2A-EGFP, and H2B-APEX2-P2A-EGFP were synthesized by Genscript and subcloned into pAAV-EF1a-DIO-WPRE-hGH vector (a gift from Dr. Karl Deisseroth, Addgene plasmid #20297) using restriction enzymes AscI and NheI. DIO abbreviation refers to the double-floxed inverted open reading frame. APEX2 sequence was cloned based on pcDNA3-APEX2-NES (a gift from Dr. Alice Ting, Addgene plasmid #49386[12]). APEX AAVs was packaged into adeno-associated virus serotype 1 by the University of North Carolina (UNC) vector core service or Vigene Biosciences (Rockville, MD. USA). AAV8.hSyn.DIO.Cherry was a gift from Dr. Bryan Roth (Cat. No. 50459-AAV8, Addgene). AAV1.CBA.DIO.hM3Dq-mCherry.WPRE was packaged by Vigene Biosciences (Rockville, MD, USA).

**Mouse strains and genotyping**. Animals were handled according to protocols approved by the Northwestern University Animal Care and Use Committee.

Weanling and young adult male and female mice were used in this study. Drd1Cre (262Gsat/Mmcd) and A2aCre(KG139Gsat)[55] were obtained from Mutant Mouse Regional Resource Center (MMRRC) at the University of California, Davis. C57BL/6 mice used for breeding and backcrossing were acquired from Charles River (Wilmington, MA). All mice were group-housed in a humidity-controlled, ambient temperature facility, with standard feeding, 12 h light-dark cycle, and enrichment procedures. Littermates were randomly assigned to conditions. All animals were genotyped according to the MMRRC strain-specific primers and protocols using GoTaq Green PCR master mix (Cat. No. M712, Promega Corporation, Madison, WI, USA).

**Stereotactic injections**. Conditional expression of APEX, hM3Dq, and reporters in Cre+ neurons was achieved by recombinant adeno-associated viral transduction encoding a double-floxed inverted open reading frame (DIO) of target genes, as described previously[49,50]. For neonatal AAV delivery, P3-6 mice were cryoanesthetized and were placed on a cooling pad. For all APEX constructs, 400 nl of AAV were delivered using an UltraMicroPump (World Precision Instruments, Sarasota, FL). Dorsal striatum was targeted in neonates by directing the needle +0.1 mm anterior of bregma, ±0.3 mm from midline, and 1.8–2.0 mm ventral to skin surface. Following the procedure, pups were warmed on a heating pad and returned to home cages, with approved post procedure monitoring. AAVs were diluted to the final titers using Gibco PBS pH 7.4 (AAV1.EF1a.DIO.APEX-NES-P2A-EGFP, AAV1.EF1a.DIO.LCK-APEX-P2A-EGFP, AAV1.EF1a.DIO.H2B-APEX-P2A-EGFP titer ~$3 \times 10^{12}$ GC/ml). For chemogenetic experiments, AAV1.EF1a.DIO.H2B-APEX-P2A-EGFP ~$3 \times 10^{12}$ GC/ml was co-injected into the striatum with either hM3Dq or mCherry fluorophore control viruses (titer: AAV1.CBA.DIO.hM3Dq-mCherry.WPRE ~$4.3 \times 10^{12}$ GC/ml, and AAV8.hSyn.DIO.mCherry ~$5.25 \times 10^{12}$ GC/ml). P35-P70 animals were used for histology, western blots, and proteomics experiments. Drd1Cre-negative littermates, which also received one of the three APEX AAVs, were used as a negative enrichment control group for proteomics, because Drd1Cre-negative littermates do not express DIO. APEX.

**Acute slice preparation and electrophysiology**. Acute slice preparation was adapted from previously published protocols[56,57]. Animals were anesthetized with isoflurane and perfused with ice-cold ACSF containing (in mM): 127 NaCl, 25 NaHCO3, 1.25 H2Na2PO4 monobasic, 25 D-Glucose, 2.5 KCl, 1 MgCl2, and 2 CaCl2 (osmolarity ~310 mOsm/L). Animals were decapitated, and the brain was immediately removed and submerged in ice-cold ACSF. Tissue was blocked using a 4% agar block and transferred into a slice chamber containing ice-cold ACSF bubbled with 95%O2/5%CO2. Bilateral coronal slices (300 μm) were cut on a Leica VT1000 S slicer. Slices were cut in lateral-medial direction and transferred into a holding chamber containing pre-warmed (34 °C) and oxygenated. Slices were incubated at 34 °C for 15 min and recovered at RT for 30 min.

Slices were transferred into a recording chamber perfused with oxygenated ACSF at a flow rate of 2–4 mL/min at RT. Whole-cell patch-clamp recordings were conducted on SPNs. Patch pipettes with ~5–8 MΩ resistance were filled with internal solution containing (in mM): 115 K-Gluconate, 20 KCl, 4 MgCl2, 10 HEPES, 4 Mg-ATP, 0.3 Na-GTP, 7 Phosphocreatine (disodium salt hydrate), and 0.1 EGTA (in KOH) (pH 7.2, osmolarity 290 mOsm). Excitability experiments were conducted in current clamp mode, where holding current was set to hold cells at approximately −70 mV. Recordings were made using a 700B amplifier (Axon Instruments, Union City, CA); data were sampled at 10 kHz and filtered at 4 kHz with a MATLAB-based acquisition script (MathWorks, Natick, MA). Series and input resistance were monitored using a 200 ms, −5 pA pulse at the end of every sweep. Acquisition intervals were 20 s long; sweeps were 5 s long with 250 ms long current injections after a 200 ms long delay.

**Quantification and statistical analysis for electrophysiology data**. Offline analysis of electrophysiology was performed using IgorPro (Wavemetrics, Portland, OR). Action potential shape analysis was performed using a MATLAB-based analysis script. Sex and age were balanced across groups. Statistical analysis was performed using GraphPad Prism software (GraphPad, LaJolla, CA). Group data were expressed as group means ± SEM. Multiple group comparisons were done using one-way ANOVA with Tukey's post hoc comparison. Adjusted $p$ value < 0.05 was considered statistically significant.

**Tissue processing, immunohistochemistry, and analysis**. Mice were deeply anaesthetized with isoflurane and transcardially perfused with 4% PFA in 0.1 M phosphate buffered saline (PBS). Brains were post-fixed for 1–5 days and washed in PBS, prior to sectioning at 50–80 μm on a vibratome (Leica Biosystems). To verify APEX activity and subcellular localization, tissue sections were incubated in PBS containing 500 μM BP (Cat. No. LS-3500, Iris Biotech) for 30 min and treated with PBS containing 0.03% H2O2 for 1 min. The reaction was quenched 3× with PBS containing 10 mM NaN3 and 10 mM sodium ascorbate. For other immunostaining, no biotinylation was performed. Sections were incubated with primary antibody with 0.2% Triton X-100 5% bovine serum albumin (BSA, Cat. No. A3059, Sigma, MO, USA) in PBS for 24–48 h at 4 °C. On the following day, tissues were rinsed three times with PBS, reacted with secondary antibody for 2 h at RT (1:500 in PBS

0.1% Triton X-100), rinsed again for three times. Sections were mounted dried on Superfrost Plus slides (Thermo Fisher, Waltham, MA), air dried, and cover slipped under glycerol:TBS (9:1) with Hoechst 33342 (2.5 µg/ml, Thermo Fisher Scientific). Primary antibodies used in the study were mouse anti-FLAG (1:1,000; Cat. No. A00187-200, Genscript, NJ, USA), chicken anti-GFP (1:2,000; Cat. No. AB13970, Abcam, Cambridge, UK), rabbit anti-c-Fos (1:10,000; Cat. No. 226003, Synaptic Systems, Goettingen, Germany), rabbit anti-JunB (C37F9) (1:5,000) (Cat. No. 3753 S, Cell Signaling, Danvers, MA). Alexa Fluor 488/594/647-conjugated secondary antibodies against mouse, rabbit, chicken and/or Alexa Fluor 647-conjugated streptavidin (Life Technologies, Carlsbad, CA) were diluted to 1:500 for all secondary antibody staining steps. Whole sections were imaged with an Olympus VS120 slide scanning microscope (Olympus Scientific Solutions Americas, Waltham, MA). Confocal images were acquired with a Leica SP5 or SP8 confocal microscope (Leica Microsystems).

ImageJ (FIJI)[58,59] was used for APEX labeling quantification. For calculating the ratio of fluorescent signal inside vs outside the cell body, somata masks were manually drawn for each confocal image frame in the GFP channel. The same mask was applied to streptavidin channel. For vertical line scan analysis, random cells across multiple confocal images were selected, fluorescent intensity for GFP and streptavidin were normalized to the corresponding maximum intensity within a given line scan and averaged across ten cells.

For validating DREADD constructs, animals expressing hM3Dq-mCherry or mCherry reporter control in striatal Drd1[Cre+] neurons were administered 3 mg/kg CNO (i.p.) and placed back into their home cage. At each time point post CNO injection (1 h, 2 h, 4 h), animals were anesthetized, and perfused with fixatives. Immunostaining was performed as indicated above. Images used in colocalization quantification were acquired under identical microscope settings. Depth-matched z-stacks of 2 µm-thick optical sections were analyzed in ImageJ (FIJI). For colocalization, the same threshold was applied for subtracting background immunofluorescence. GFP (APEX) or mCherry signal was used to localize cell bodies of hM3Dq-mCherry or mCherry expressing neurons. cFos signal was identified independently from GFP or mCherry signal by an experimenter blind to the conditions. Statistical analysis was performed using GraphPad Prism software (GraphPad, LaJolla, CA). Group data are expressed as group means ± SEM. Multiple group comparisons were done using one-way ANOVAs with Tukey's correction. Adjusted $p < 0.05$ was considered statistically significant.

**Evaluating expression variability and analysis of minimal labeling depth.** 250 µM thick acute slices were prepared and labeled as described above with 250 µM PT supplementing ACSF. PT was used here instead of BP, because PT-labeled proteins can be detected by a small-molecule clickable dye. A small molecule detection reagent has a greater tissue penetration compared to streptavidin-BP. After labeling, slices were fixed in 4% PFA in PBS at 4 °C overnight. Slices were rinsed with 1× PBS three times. Labeled proteins were probed using 2.5 µM picolyl-azide-594 in a click-reaction mixture (0.5% triton-X100, 50 µM CuSO₄, 250 µM BTTAA, 5 mM Na ascorbate in PBS) for 1 or 2 h, as noted. Slices were washed three times with PBS for 5 min each before mounted on super frosted glass slides with 250 µm spacers before cover slipping.

For imaging, a single-objective-based scanned oblique plane illumination (SOPi) microscope was utilized for fluorescence imaging through the tissue depth[25]. SOPi was configured to perform direct light-sheet fluorescence imaging of a 45° oblique plane in the sample. A galvanometer scanner (GVS211, Thorlabs) was placed in the back focal plane of the sample facing objective to provide a tilt-invariant lateral scan of the illumination light-sheet[60]. In the illumination path, two lasers (MGL-III-532-100mW, MBL-III-473-50mW, DragonLasers) were co-aligned with a dichroic mirror (FF495-Di03) to enable excitation of green and red fluorophores. The combined beams were rapidly scanned through a galvanometer scanner (GVS201, Thorlabs) and a scan lens (AC-508-150-A-ML, Thorlabs) to create a light-sheet[61]. The imaging arm consisted of a modified SOPi microscope with three 60×, 1.0 numerical aperture (NA) water immersion microscope objective lenses, each paired with standard tube lenses (SWTLU-C, Olympus). The light-sheet fluorescence images were captured using corresponding fluorescence filters (MF525-39, MF620-52, Thorlabs) on a CMOS camera (GS3-U3-23S6M-C, Flir). This arrangement resulted in an effective pixel size of 175.8 nm and overall system NA of 0.6. With 1900 by 1200 pixels, each camera frame corresponded to ~335 × 210 µm². Here, the smaller dimension i.e., 210 µm is along the 45° oblique depth resulting into ~150 µm vertical depth coverage. This microscopy system was controlled with a custom written python program to synchronize the lateral scan in the light-sheet and camera acquisition trigger[26].

**Synthesis of propargyl tyramine.** In a 20 mL vial, 4-(2-aminoethyl) phenol (0.3 g, 2.2 mmol), 2,5-dioxopyrrolidin-1-yl 3-(prop-2-yn-1-yloxy) propanoate (0.5 g, 2.2 mmol), DIPEA (1.94 mL, 11.1 mmol) in DCM (10 mL) was stirred at room temperature for 3 h. On completion the reaction was further diluted with CH₂Cl₂ (5 mL) and washed with HCl (2 N, 2 × 10 mL) and brine (15 mL). The organic portion was evaporated to yield a light brown residue. The residue was purified via Biotage (10:1 CH₂Cl₂/MeOH; 10 g column). Collected fractions: tube 1 impure, tube 2 and 3 pure to yield a light brown-colored oil. ¹H NMR (500 MHz, CDCl₃) δ 7.09–6.97 (m, 2H), 6.85–6.68 (m, 2H), 6.13 (m, 2H), 4.08 (d, $J$ = 2.4 Hz, 2H), 3.71 (t, $J$ = 5.7 Hz, 2H), 3.48 (td, $J$ = 7.0, 5.8 Hz, 2H), 2.72 (t, $J$ = 6.9 Hz, 2H), 2.54–2.32 (m, 3H);

¹³C NMR (126 MHz, CDCl₃) δ 171.4, 154.7, 130.4, 129.8, 115.5, 79.1, 75.0, 65.9, 58.3, 40.8, 36.9, 34.7; MS: C₁₄H₁₇NO₃ $m/z$ 248.2 [M + H]⁺. ¹H and ¹³C NMR spectra are included in Supplementary Fig. 12.

**Open-field locomotion.** During the active phase of the circadian cycle, animals (>P35) expressing hM3Dq-mCherry or mCherry reporter control in the dorsal striatum of Drd1[Cre+] neurons were habituated in an 48 × 48 cm open-field arena for 5 min. Three mg/kg CNO (i.p.) was administered 30 min prior to behavioral assay. Animals were placed back the arena and their position were recorded for 10 min at 30 frames per second. Toxtrac[62] was used to track the animal's position, defined by its body center position, and quantify the distance travelled in each session. Group data are expressed as group means ± SEM. Welch's unpaired $t$-test was used to test for significance.

**Transmission electron microscopy (TEM).** Transcardial perfusion was performed as described above. For TEM specimen preparation, perfusion was performed with ice-cold PBS, followed by ice-cold 2% glutaraldehyde and 2% PFA in PBS. The brain was post-fixed in the same fixative overnight at 4 °C. 100 µm coronal brain slices were prepared with a Leica VT1000 vibratome. To selectively label APEX-containing cellular structures, slices were incubated with 3,3′-Diaminobenzidine (DAB) with metal enhancer (Cat. No. D0426, Sigma). Briefly, DAB solution (0.25 mg/ml DAB, 0.1 mg/ml CoCl₂, 0.15 mg/ml H₂O₂) was prepared by dissolving DAB and hydrogen peroxide tablets each in 5 ml PBS. Solutions were mixed 1:1 immediately before use. Brain slices were incubated in DAB solution from 2 to 5 min until APEX-positive cell bodies were visible. After DAB precipitation, slices were washed several times with 0.05 M sodium phosphate buffer (PB), and then processed for TEM with two exchanges of the primary fixative that consisted of 2.5% glutaraldehyde, 2% PFA in 0.1 M PB. The brain slices were again washed 3× with the buffer followed by a secondary fixation in 1.5% osmium tetroxide (aqueous). Samples were washed 3× with DI water before beginning an acetone dehydration series. All of the preceding steps up were carried out in Pelco Biowave Microwave with Cold Spot and vacuum. EMBed 812 embedding media by EMS was gradually infiltrated with acetone for flat embedding. The selected ROI was cut out and mounted on a blank stub for sectioning. 90 nm thin sections were collected on copper grids using a Leica Ultracut S ultramicrotome and DiATOME 45° diamond knife. Images were acquired at 100 kV on a 1230 JEOL TEM and Gatan Orius camera with Digital Micrograph software.

**Ex vivo biotinylation for proteomics studies.** Acute slices were prepared as described above. Viral expression in 250 µm-thick striatal slices was confirmed by NightSea Dual FP flashlight. For ex vivo biotinylation, slices were incubated in carbogenated ACSF with 500 µM BP at RT for 1 h. They were briefly rinsed in ACSF and then transferred into ACSF containing 0.03% H₂O₂ for 1 min. The reaction was quenched by transferring slices to ASCF containing 10 mM NaN₃ and 10 mM sodium ascorbate. Fluorescent-positive regions were dissected in ice-cold ACSF under a fluorescent dissection microscope (NIGHTSEA, Lexington, MA, USA) and transferred to 1.5 ml polypropylene tubes. Tissues were frozen and stored at −80 °C for further processing.

**Protein extraction, streptavidin enrichment, and on-bead trypsin digestion.** Total protein was extracted by sonication in 400 µl lysis buffer (1% sodium dodecyl sulfate (SDS), 125 mM triethylammonium bicarbonate (TEAB), 75 mM NaCl, and Halt™ protease and phosphatase inhibitors). Lysates were cleared by centrifugation at 12,000 $g$ for 15 min at 4 °C. Supernatant was transferred to a new tube and used for subsequent procedure. Total protein (1 µl of samples were diluted to 100 µl with water) was estimated using microBCA assay according to the manufacturer instructions (Cat. No. 23235, Thermo Fisher).

For the experiment in Figs. 2–3 (the proteome across three APEX constructs), 300 µg of brain lysates (in 200 µl) were reduced with 20 µl 200 mM dithiothreitol (DTT) for 1 h and alkylated with 60 µl 200 mM iodoacetamide (IAA) in the dark for 45 min at 37 °C with shaking. Streptavidin magnetic beads (300 µl) (Cat. No. 88816, Thermo Fisher) were prewashed with 1 ml no-SDS lysis buffer and incubated with 240 µl of reduced and alkylated lysates for 60 min at RT with shaking. Enriched beads were washed twice with 1 ml no-SDS lysis buffer, 1 ml 1 M KCl, and five times with 1 ml 100 mM TEAB buffer. Washed beads were digested with trypsin/LysC solution (~6 µg in 100 µl 100 mM TEAB) overnight at 37 °C. Digested supernatant was collected. Beads were rinsed with 50 µl 100 mM TEAB. Supernatant was combined. Trace amounts of magnetic beads were removed twice by magnetization and tube changes. 10 µl from each sample was saved for Pierce fluorometric peptide quantification assay (Cat. No. 23290, Thermo Fisher), and 16 µl was taken from each sample and was combined to make an average reference sample for TMT batch effect correction. Peptides were frozen and dried in a vacuum concentrator before TMT labeling.

For experiments in Figs. 4–5 (Drd1[Cre] vs A2a[Cre], and chemogenetic activation/ DREADDs), 300 µg of brain lysates (in 250 µl) were reduced with 20 µl 200 mM DTT for 1 h and alkylated with 60 µl 200 mM IAA 100 mM TEAB in the dark for 45 min at 37 °C with shaking. Streptavidin magnetic beads (150 µl) were prewashed with 1 ml no-SDS lysis buffer and incubated with 250 µl of reduced and alkylated

lysates for 60 min at RT with shaking. Enriched beads were washed twice with 1 ml no-SDS lysis buffer, 1 ml 1 M KCl, and five times with 1 ml 100 mM TEAB buffer. Washed beads were digested with trypsin solution (~4 µg in 150 µl 100 mM TEAB) overnight at 37 °C. Digested supernatant was collected. Beads were rinsed with 50 µl 100 mM TEAB. Supernatants were combined. Trace amounts of magnetic beads were removed twice by magnetization and tube changes. 10 µl from each sample was saved for Pierce fluorometric peptide quantification assay. Peptides were frozen and dried in a vacuum concentrator before TMT labeling. To make reference channel samples for TMT batch correction in Fig. 5, a separate set of H2B-hM3Dq, H2B-mCherry, and H2B-alone samples were prepared and pooled to make two 900 µg samples which were enriched with 400 µl of streptavidin beads, followed by the same bead washing and tryptic digestion (~6 µg trypsin in 200 µl + 50 µl rinse). Eluted peptides were pooled and dried in a vacuum concentrator.

**TMT labeling, fractionation, and desalting**. For the experiment in Figs. 2–3 (the proteome across three APEX constructs), dried peptide samples were reconstituted in 20 µl 100 mM TEAB and sonicated for 15 min at RT. TMT labeling protocol was performed similar to Zecha et al. (20 µl peptides + 5 µl 59 mM TMTzero)[63]. Briefly, one set of 0.8 mg TMT 10plex reagents was warmed up to room temperature and dissolved in 41 µl Optima LC/MS-grade acetonitrile (ACN). 5 µl of ~56 mM TMT reagents was added to 20 µl reconstituted peptide according to the experimental design in Supplementary Fig 5a. Labeling was performed at RT for 1 hr with shaking at 400 rpm. The reaction was quenched by adding 2.2 µl 5% hydroxylamine/100 mM TEAB at RT for 15 min with shaking. For a shotgun experiment, 12 µl of each sample was mixed equally, acidified to pH < 3 with formic acid (FA) and dried in a speed vacuum concentrator. TMT mixture was resuspended in 50 µl buffer A (LCMS water 0.1% FA) and desalted using Pierce C18 spin tip (Cat. No. 84850, Thermo Fisher). All centrifugation steps were performed at 1000 g for 1 min. Spin tips were activated twice using 20 µl 80% ACN 0.1% FA and equilibrated twice using 20 µl buffer A. Samples were loaded ten times. Spin tips were washed twice with 20 µl buffer A. Peptides were eluted with 40 µl 80% ACN 0.1% FA and dried in a vacuum concentrator and stored at −80 °C until MS data collection. The remaining samples (12 µl) were mixed equally and dried in a vacuum concentrator for high pH reverse-phase fractionation (Cat. No. 84868, Thermo Fisher). Briefly, samples were resuspended and sonicated for 10 min in 300 µl buffer A (LC MS water 0.1% FA) supplemented with 1 µl FA. Sample acidity was verified by pH papers. Resin was packed by centrifugation at 5000 g for 2 min, activated twice with 300 µl ACN, and conditioned twice with 300 µl buffer A. Peptides were loaded five times by centrifugation at 3000 g for 2 min. Column was first washed with 300 µl water, and eluted by increasing percentage of acetonitrile in 0.1% trie-thylamine solution according to the manufacturer instructions (5%, 10%, 12.5%, 15%, 17.5%, 20%, 22.5%, 25%, and 50% ACN). All fractions were dried in a vacuum concentrator.

For experiments in Figs. 4–5 (Drd1^Cre vs A2a^Cre, and chemogenetic activation), dried peptide samples were reconstituted in 20 µl 100 mM TEAB and sonicated for 15 min at RT. TMT labeling protocol was performed as described according to the experimental design in Supplementary Figs. 7,9. TMTPro 16plex reagent (Cat. No. A44521, Thermo Fisher) and TMT 11plex reagent (Cat. No. A37725, Thermo Fisher) were used for experiments in Figs. 4–5, respectively. Six µl TMTPro 16plex (~59 mM, 0.5 mg reconstituted in 20 µl ACN) and 5 µl of TMT 11plex (~56 mM, 0.8 mg reconstituted in 41 µl ACN) were used, respectively. Labeling was performed at RT for 90 min with shaking at 400 rpm. The reaction was quenched by adding 3 µl 5% hydroxylamine/100 mM TEAB at RT for 15 min with shaking. Samples (20 µl each) were mixed equally, dried in a dried in a vacuum concentrator, and fractionated as described above.

5–50% ACN fractions were used for LC MS/MS analysis for experiments in Figs. 3 and 5, and 10–25% ACN fractions for experiments in Fig. 4.

**Western blotting**. Tissue lysis and biotinylated protein enrichment procedure were described in method section above. Protein lysate inputs or flow through fractions were mixed with 6× Laemmli loading buffer and heated to 90–95 °C for 10 min. For eluting biotinylated protein off the beads, washed beads were mixed with 20 µl of 2× Laemmli buffer containing 25 mM TEAB, 75 mM NaCl, and 20 mM biotin. Beads were heated to 90–95 °C for 10 min. Proteins were separated in 4–20% gradient gels (Cat. No. 4561096, Biorad, CA, USA) and transferred to nitrocellulose membrane (Cat. No. 926-31090, LI-COR, NE, USA). Blots were briefly rinsed with TBS. For detection of biotinylated proteins, blots were incubated in TBST (0.1% Tween-20) containing streptavidin CW800 (1:10,000, Cat. No. 926-32230, LI-COR) for 1 h at RT. Blots were washed three times with TBST for 10 min each. Total protein was detected using REVERT 700 according to the manufacturer instructions. Blots were scanned using a LI-COR Odyssey CLx scanner. All quantification was performed using LI-COR Image Studio version 5.2.

**Cell culture and biochemical subcellular fractionation**. HEK293T cells were obtained from ATCC and maintained in complete Dulbecco's Modified Eagle Medium (DMEM), supplemented with 10% fetal bovine serum (FBS) and 1% penicillin-streptomycin in 37 °C/5%CO2 incubator (Cat. No. 11965118, 10437028, 15140122, Thermo Fisher).

For biochemical subcellular fractionation experiment, cells were plated at 75% confluency in 10 cm dish and transfected 4 h later. Cells were transfected using

linear 25k polyethylenimine (Cat. No. 23966-1, Polysciences, Warrington, PA). 6 µg APEX-containing plasmids ± 2 µg EF1a-Cre plasmid were diluted and vortexed in 1 ml OptiMEM (Cat. No. 31985088, Thermo Fisher). 36 µg PEI (1 mg/ml stock) was added. DNA-PEI solution was vortexed and incubated at RT for 15 min before adding dropwise to cell culture dishes. After 16 h post transfection, media was changed to complete DMEM with 10% FBS supplemented with 250 µM BP for 1 h. Cells were washed with 5 ml DPBS (Cat. No. 14190250, Thermo Fisher). Biotinylation was induced by adding 2 ml 0.03% H2O2 DPBS for 1 min. Cells were washed with 5 ml quench solution. Cells were lifted in 1 ml quenching buffer and centrifuged at 300 g for 3 min (~500 µl volume of HEK293T pellet). Pierce subcellular fractionation kit was used to enrich nuclear, cytosolic and membrane proteins (Cat. No. 78840, Thermo Fisher). HEK293T cell pellets were lysed in 500 µl CEB buffer. Pellets were gently dispersed by inverting the tubes and incubated on ice for 10 min. Total protein extracts were centrifuged at 500 g for 5 min at 4 °C. Cytoplasmic extract supernatant was collected. Next, the pellets were gently resuspended in 500 µl MEB buffer and incubated on ice for 10 min to extract crude membrane fraction. Extracts were centrifuged at 3000 g for 5 min. Crude membrane extract supernatant was collected. Lastly, the pellets were briefly vortexed in 250 µl NEB buffer and incubated on ice for 30 min to extract nuclear proteins. Soluble nuclear fractions were obtained by collecting supernatant after centrifugation at 5000 g for 5 min. Pellet was resuspended in 120 µl NEB supplemented with 10 mM CaCl2 and 2.5 µl MNase (100 units/µl) at RT for 15 min. Chromatin-bound extract supernatant was collected after centrifugation at 16,000 g for 5 min. Both nuclear extracts were combined. BCA assay was used to estimate total protein. Equal amounts of proteins (~16 µg) were used in SDS-PAGE and western blot analysis (15% gels).

For imaging, cells were plated at 75% confluency in poly-L-lysine (0.01%, Cat. No. P8920, Sigma) coated 96-well plate and transfected 4 h later. Transfection was carried out as described above with 150 ng APEX DNA and 50 ng EF1a-Cre plasmids. Biotinylation was performed as described above. After biotinylation, cells were fixed with 4% PFA in PBS for 10 min and washed three times with PBS. Biotinylated proteins were detected with 1:2000 streptavidin-AF647 (in PBST, 20 min at RT). Cells were washed three times with PBS. Nuclei were stained with 100 ng/ml Hoescht 33342 in PBS. Cells were imaged with BioTek Lionheart FX under the same setting in PBS.

**Mass spectrometry data acquisition and raw data processing**. TMT labeled peptides were resuspended in 2% acetonitrile/0.1% FA, and ~1 µg was loaded onto a heated PepMap RSLC C18 2 µm, 100 angstrom, 75 µm × 50 cm column (ThermoScientific) and eluted over 180 min gradients optimized for each high pH reverse-phase fraction (Supplementary table 1). Sample eluate was electrosprayed (2000 V) into a Thermo Scientific Orbitrap Eclipse mass spectrometer for analysis. MS1 spectra were acquired at a resolving power of 120,000. MS2 spectra were acquired in the Ion Trap with CID (35%) in centroid mode. Real-time search (RTS) (max search time = 34 s; max missed cleavages = 1; Xcorr = 1; dCn = 0.1; ppm = 5) was used to select ions for SPS for MS3. MS3 spectra were acquired in the Orbitrap with HCD (60%) with an isolation window = 0.7 m/z and a resolving power of 60,000, and a max injection time of 400 ms.

Raw MS files were processed in Proteome Discoverer version 2.4 (Thermo Scientific, Waltham, MA). MS spectra were searched against the Mus musculus Uniprot/SwissProt database. SEQUEST search engine was used (enzyme=trypsin, max. missed cleavage = 4, min. peptide length = 6, precursor tolerance = 10 ppm). Static modifications include acetylation (N-term, +42.011 Da), Met-loss (N-term, −131.040 Da), Met-loss+Acetyl (N-term, −89.030 Da), and TMT labeling (N-term and K, +229.163 Da for TMT10/11, or +304.207 Da for TMTpro16). Dynamic modifications include oxidation (M, +15.995 Da). PSMs were filtered by the Percolator node (max Delta Cn = 0.05, target FDR (strict) = 0.01, and target FDR (relaxed) = 0.05). Proteins were identified with a minimum of one unique peptide and protein-level combined q values < 0.05. Reporter ion quantification was based on corrected S/N values with the following settings: integration tolerance = 20 ppm, method = most confident centroid, co-isolation threshold = 70, and SPS mass matches = 65. PSMs results from Proteome Discoverer were exported for analysis in MSstatsTMT R package (version 1.7.3).

**Statistical analysis with MSstatsTMT**. PSMs was exported from Proteome Discoverer and converted into MSstatsTMT-compatible format using PDtoMS-statsTMTFormat function. PSMs were filtered with a co-isolation threshold = 70 and peptide percolator q value < 0.01. For protein quantifications, only unique peptides were used. In addition, only proteins with a minimum of two unique PSMs were quantified (e.g., proteins with one unique peptide must have at least two PSMs of different charges to be considered for summarization in the MSstatsTMT). Therefore, not all identified proteins were quantified. Protein summarization was performed in MSstatsTMT with the following arguments: method = msstats, global median normalization = FALSE, reference normalization = TRUE, imputation = FALSE.

For comparison between APEX constructs in Figs. 2–3, to reflect enrichment differences across the APEX constructs at the protein level, global median normalization was not performed. It was necessary to perform reference sample normalization to correct for TMT batch effects[64]. MSstatsTMT implements this correction protein-by-protein. Gene ontology for cellular component was imported

to R from Proteome Discoverer. Pairwise differential expression analysis was performed using moderated $t$-tests with Benjamini–Hochberg (BH) multiple hypothesis correction. $q$ values (adjusted $p$ values) < 0.05 (5% FDR) were considered as statistically significant.

For comparison between Drd1$^{Cre+}$ and A2a$^{Cre+}$ in Fig. 4, summarization was performed as described above with global median normalization = TRUE (at the peptide level). Nonspecific binding entries were removed using a list compiled from experiments in Fig. 5, including astrocyte-enriched proteins. A list of astrocyte-enriched proteins was generated from DropViz (http://dropviz.org/[2]) with the following setting (STR_astrocyte_Gja1[#4] vs rest of striatum, minimum fold ratio = 9.5, maximum $p$ value exponent = −100 AND min mean log amount in target = 2.5, and max mean log amount in Comp = 4). A complete list of contaminants can be found in the Supplementary Data 4 tab 2. This contaminant list includes (i) proteins that are non-specifically enriched in the comparison between Cre+ and Cre- samples in Fig. 5 (i.e., proteins that did not pass the moderated $t$-test cutoff: 5% FDR and log2FC > 0), (ii) mouse keratin, and (iii) astrocyte-enriched proteins. In addition to nonspecific binding contaminants, we removed some abundant astrocyte-enriched proteins because in this experiment the goal is to compare two neuronal subtype proteomes. After removing nonspecific contaminants, retained proteins were median normalized. The H2B-enriched protein list was generated by comparison H2B and NES samples, separately for Drd1$^{Cre}$ and A2a$^{Cre}$ samples. Proteins that passed 5% FDR cutoff and log2FC > 0 were annotated as H2B-enriched. For comparison between Drd1$^{Cre}$ and A2a$^{Cre}$, NES and H2B samples were tested, separately (i.e., A2aCre_NES – D1Cre_NES, and A2aCre_H2B – D1Cre_H2B). $p$ values were used in the volcano plot for this comparison.

For chemogenetic experiments in Fig. 5, nonspecific binding was removed by moderated $t$-tests between Cre+ and Cre- samples (5% FDR and log2FC > 0). After removing nonspecific binding proteins, H2B-enriched protein list was generated by comparison between H2B-hM3Dq/mCherry and NES samples. Proteins that pass 5% FDR cutoff are annotated as H2B enriched, otherwise, non-H2B enriched. After assigning subcellular annotation, each protein set was median normalized at peptide and protein level. Moderated $t$-tests were used to assess the statistical significance between hM3Dq and mCherry samples. All statistical comparisons are included in the supplementary materials.

Multidimensional scaling (MDS) was done using the plotMDS function in edgeR version 3.30.3[65] using the default setting (top 500 proteins). Entries with missing values were omitted in this clustering. Coefficient of variations were calculated on log2 protein intensities. Multi-scatter plots were generated using pairs.panels function in psych version 2.0.7. Heatmap was generated using ComplexHeatmap R package[66] with a default setting from differentially regulated proteins after a nominal cutoff ($p$ value < 0.3 for Supplementary Fig. 8, and $q$ value < 0.05 for Fig. 5).

For gene ontology analysis, DAVID v6.8[67] (https://david.ncifcrf.gov/) was used with identifier = 'UNIPROT_ACCESSION'. All identified proteins within each dataset were used as background. In Fig. 3, H2B-enriched and LCK-enriched lists were used as input. In Fig. 4, H2B-enriched proteins were used as input. In Fig. 5, NES-enriched and H2B-enriched lists were used as input. −log10(FDR) was plotted for top GO terms.

**Correlation analysis between proteins and transcripts**. Known differentially expressed transcripts between dSPNs and iSPNs were generated from previously reported TRAPseq and scRNAseq data (Heiman et. al. Cell[20], Gokce et. al. Cell Reports[21], and Montalban et. al. BioRxiv[42]). First, genes with adjusted $p$ values < 0.05 and log2FC > 1 (Drd1-enriched) or log2FC < −1 (Drd2-enriched) were taken from Heiman et al. Supplementary Table S2. Second, genes that are associated with 'D1-MSN' and 'D2-MSN' gene clusters found in Gokce et al. Supplementary Table 6 were considered as differentially expressed genes. Lastly, genes with 'D1D2inDS' adjusted $p$ values < 0.05 and log2FC > 1 (Drd1-enriched) or log2FC < −1 (Drd2-enriched) were considered as differentially expressed from Montalban et al. Supplementary Table S2. The three lists were merged. Non-redundant gene name entries were mapped to the Uniprot (SwissProt) database for *Mus musculus*. Scatter plots of protein and transcripts log2FC were generated only for proteins that were quantified by MSstatsTMT in our dataset.

**Network analysis**. JunB network information was obtained from STRING-DB v11[68] (https://string-db.org/) and HuRI database[69] (http://www.interactome-atlas.org/). Interactions were mapped to the mouse Swissprot accession. Only nodes and edges derived from proteins identified in the dataset were included. Network figures were created using Cytoscape (v.3.8.2)[70] with nodes corresponding to gene names. Node shapes correspond to molecular function. Node color is log2FC (hM3Dq – mCherry) directly calculated from Proteome Discoverer reporter ion intensities, respectively.

**Reporting summary**. Further information on research design is available in the Nature Research Reporting Summary linked to this article.

**Data availability**
The raw MS data generated in this study have been deposited in the PRIDE database under accession code PXD022335. Analyzed data generated in this study are provided in the Supplementary Information and Source Data file. The reference number for the mouse SwissProt database used in this study is 000000589 [https://www.uniprot.org/uniprot/?query=proteome:UP000000589%20reviewed:yes]. For network analysis, STRING-DB v11 (https://string-db.org/) and HuRI (http://www.interactome-atlas.org/) were used. All datasets and plasmids generated in this study are available from the corresponding author on reasonable request. Source Data are provided. Source data are provided with this paper.

**Code availability**
All analysis code is available on Github at https://github.com/KozorovitskiyLaboratory/proteomics_APEX.

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

## Acknowledgements

The authors thank the reviewers for helpful feedback, which improved the manuscript. The authors are grateful to Lindsey Butler for mouse colony management, Northwestern Biological Imaging Facility and Dr. Tiffany Schmidt for confocal microscope access, Northwestern University BioCryo Facility (Charlene Wilke and Dr. Reiner Bleher) for TEM sample preparation and microscope, Northwestern High Throughput Analysis laboratory for the microplate reader, and Northwestern University Center for Molecular Innovation and Drug Discovery (Dr. Atul Dilip Jain and Dr. Gary Schultz) for compound synthesis. Some schematics were created with BioRender.com. This work was supported by the NSF CAREER Award 1846234, NIMH R56MH113923, NINDS R01NS107539, NIMH R01MH117111, the Beckman Young Investigator Award, Searle Scholar Award, Rita Allen Foundation Scholar Award, and Sloan Research Fellowship (all Y.K.), and NIMH R01MH118497 (M.L.M.). V.D. is a predoctoral fellow of the American Heart Association (19PRE34380056) and as an affiliate fellow of the NIH 2T32GM15538.

## Author contributions

V.D., M.L.M., and Y.K. designed the study. V.D. carried out most experiments and analyses in the study. V.D. created new plasmids, viruses, performed MS sample preparation and contributed to experimental analyses. M.L.M. and R.B.S. performed MS sample preparation, data acquisition and analyses. G.S. performed electrophysiology experiments and analyses. M.K. performed light-sheet microscopy experiments and analyses. V.D., M.L.M., and Y.K. wrote the paper, with feedback from all authors.

## Competing interests

The authors declare no competing interests.
