## [Peer Review File · Nature Communications]

REVIEWER COMMENTS

Reviewer #1 (Remarks to the Author):

This manuscript from Dumrongprechachan et al describes an interesting proximity labeling approach for cell type- and compartment-specific proteomics in the mouse brain. The authors use a combination of the *Drd1*-cre mouse line with cre-dependent adeno-associated viruses to express a proximity labeling reagent, APEX, in different subcellular compartments of DRD1+ neurons. Rapid peroxide-dependent labeling was accomplished *ex vivo* on brain slices. Gel-based analysis confirmed that this labeling system was working as expected. Quantitative TMT proteomics of DRD1+ neuronal proteomes confirmed enrichment of proteins over background, some of which were known (eg *Hnrnpa2b1*) and others that were previously unknown to be enriched in DRD1+ proteomes.

Overall, I found the experiments presented in this manuscript to be interesting and well done. The text is clearly written. The authors have convincingly demonstrated that the APEX system works well in mouse brain. However, my enthusiasm for the manuscript in its current form is still limited. This is largely because there is already similar published work on cell type-selective proteomics or proximity labeling in mouse brain (PMID 29251727, 29106408, 27609886, 30674877, 33066078). This manuscript would be a major advance and distinct from this existing literature if they could demonstrate either generality across multiple cell types or applicability for dynamically changing proteomes. While these applications are provocatively suggested in the text, no experimental data are provided for their feasibility. If the authors could address this issue, in my opinion this work would be of broad interest to the community and wider field.

Major comments:

1. As mentioned above, this manuscript would be greatly strengthened if the authors could demonstrate the generality of their method. Obviously many different cre drivers are available for genetic manipulation in the brain. It would be of great interest to compare nuclear proteomes across a few different types of neurons (or even between neurons and non-neuronal cell types such as glia).
2. Another major advantage of this method could be to monitor proteome dynamics on short time scales, an application that is not easily achieved by unnatural amino acid incorporation methods due to slow metabolic labeling. What happens to *Drd1* proteomes following pharmacological or behavioral challenge?

Minor comments:

3. It would be helpful if the authors could validate at least some of their proteomic data by staining or some other orthogonal approach. Are these identified proteins actually enriched in DRD1+ neurons, or are they general nuclear neuronal markers?

Reviewer #2 (Remarks to the Author):

The manuscript delivers on providing a demonstration of the ability to enrich and identify proteins from *Drd1* positive cells and enrich for proteins from different compartments. This approach will be of interest to the neuroscience community and will enable a better understanding of the proteins involved in achieving different functions in brain circuitry. The value of the subcellular compartments is less well justified by the authors. They appear to be the first to demonstrate the overall approach. The proteomics and statistics methodology were appropriately performed and to a high standard.

There is a major issue with the description of the control for the proteomics analysis. The control is not described anywhere, except in the text as "Cre-negative". One presumes that C57BL/6 were used. However, the only mention of C57BL/6 mice is: "C57BL/6 mice used for breeding and backcrossing". It is also possible that a mock injection was done, or a vector containing EGFP, either with or without the DIO, was used as control on *Drd1*(Cre) mice. The latter may have been a better control than using C57BL/6 mice. Nothing was written to help the reader. A control (AAV8.hSyn.DIO.Cherry) is used for electrophysiology. There's no indication whether that control was

also used in the proteomics; presumably not because it would be strange to call it “Cre-negative”.

The control is important because it gets to the heart of whether this data is biologically useful. We are presented with: (a) lists of proteins enriched in Drd1+ cells and (b) lists of proteins enriched in the nucleus and membrane of Drd1+ cells. The list of Drd1+ cell enriched proteins is not particularly useful when compared to the control. The control is mainly a list of proteins that bind the beads non-specifically plus those mitochondrial proteins which are endogenously biotinylated. The list does not tell us whether these proteins are enriched in this cell type compared to any other brain cell type. Thus, the value is limited. The only immediate way to increase value would be to validate one/some of these results. Then we have the list of compartment-enriched proteins. The list helps identify some proteins, “Dock3 and Actn4”, that may be newly identified as membrane proteins. However, the immediate use of defining compartments is not obvious. Furthermore, the membrane and cytosolic compartments had no significant differences. Perhaps if the proteins changed compartment following stimulation, then there will be mechanistic value. When considering the wording of the abstract, introduction and concluding sentences, the article is sold on the value of the cell type-specific analysis, but the main deliverable is the nucleus and membrane compartment comparison.

When considering that the depth of the proteome analysis is not particularly deep (1,552 proteins. 736 quantified. They concede that fractionation prior to LC-MSMS could have helped), the overall impression is that this is a new approach that has been rushed to publication as an article of high novelty, but low immediate biological value, especially to those with interest in Drd1+ cells. The article may have immediate value to those interested in performing a similar cell type-specific analysis. Overall, I am concerned that the manuscript has low value beyond its novel approach.

Minor issues:

P2A not explained as self-cleaving element of their vector.

“SNr” abbreviation not defined.

“SPS MS3”. “SPS” not defined.

Beginning of 10th paragraph “Genetically targeted proximity labeling approaches offer information on cell-type specificity and subcellular localization.” The sentence seems repetitive and unnecessary here. Suggest deleting.

A minor, but troublesome issue is whether the streptavidin binding sites on the beads were in excess compared to the biotinylated beads. If not, the biotinylated proteins, including endogenous biotinylated proteins, would be in competition for streptavidin binding sites and this may affect any comparisons if, for example, there were more endogenous biotinylated proteins in one sample than another. Ideally, an experiment would be performed to show at what point a fixed amount of beads is able to extract maximum protein from a varying level of lysate (and then a lysate level is used below this saturation point to ensure minimal competition).

“This was not surprising because mitochondrial compartment is oxidative.” This reason is a little speculative and should be further supported if the authors insist on using it. Mitochondrial contamination could simply be due to the high abundance of mitochondrial proteins or indirect binding of mitochondrial proteins to biotinylated mitochondrial proteins (i.e. extraction of mitochondrial protein complexes).

“Conditional expression of APEX and reporters in Cre+ neurons was achieved by recombinant adeno-associated viral transduction encoding a double-floxed inverted open reading frame (DIO) of target genes, as described previously.” There should be a reference at the end of this sentence.

“double-floxed inverted open reading frame (DIO)” abbreviation appears above this in “Plasmid construction and AAV preparation” section.

It should be stated in the methods that the minimum number of peptides required for protein identification was 1. This is reported in the table but must be stated in the methods.

Having text that is 2-4 point in size, as in Fig. 3e, is not desirable. I suggest making the figure larger.

“Dock3 and Actn4” are gene names. When referring to the proteins, protein names should be used. Of course, genes names can be used, but it should be obvious that they are not the same as the protein names. Mouse gene names should be italicised, including when shown in figures (e.g. Fig. 3e).

Reviewer #3 (Remarks to the Author):

This manuscript describes a proteomics method to identify cell type and subcellular compartment specific proteomes in the mouse brain by using genetically targeted proximity biotinylation labeling employing APEX2. Specific subcellular compartments were interrogated by creating three Cre-dependent APEX variants that localized to the nucleus, to the cytosol, and to the membrane compartment by tagging with well validated sorting motifs (Histone 2B fusion, H2B; nuclear exporting sequence, NES; and membrane anchor LCK sequence, respectively) and coupling with TMT analysis. A cell type-specific component was included by use of expression of the APEX2 variants in Drd1 striatal medium spiny neurons. Overall, the authors were able to identify 1552 proteins (but see comment below) and successfully classified differential localization of 10s of proteins specific to cell type and subcellular compartment.

Overall, this is a solid study, with careful design and analysis. The manuscript is well written and the interpretation of the results is appropriate. The work expands examples of the utility of APEX2 in proximity biotinylation studies. However, the study is missing an essential element, namely a comparison of 2 or more different cell types. An obvious comparison would be with Drd2-expressing medium spiny neurons. Alternatively, comparison with cerebellar, cortical or hippocampal neurons would be informative. Only then would one know if the approach has sufficient proteomic depth to identify and quantify differentially expressed gene products, and to be comparable to scRNAseq or TRAP approaches. There is also little new biological insight provided. Some sort of drug treatment or behavioral perturbation would be needed.

Specific comments:

1. The authors show the subcellular specific localization of Cre-dependent APEX variants by microscopy. Did the authors also examine this by western blot after fractionation of nucleus, cytosol and membrane fraction from total striatal tissue.
2. The authors should provide separate lists of identified proteins in each compartment.
3. The authors should provide more details about data processing like how many minimum peptides were used for protein IDs and peptide and protein FDR in the methods section.
4. Two filter criteria were used to isolate cell type and subcellular specific proteomes 1) comparisons between Cre-positive samples and Cre-negative control samples; 2) stringent cutoff for the nuclear and membrane-enriched proteomes, using moderate t-tests. Can the authors explain about handling the proteins which are present in two places like nucleus and cytosol. Maybe these proteins are filtered out during differential analysis. This might explain not getting any different proteins in the NES-LCK comparison.
5. Fig 1F – why does the protein stain look identical in all conditions? Are these abundant endogenously biotinylated proteins found in all samples.
6. Ex vivo slice dissection – how consistent was the fluorescent signal through a section/slice?
7. TMT methods – was there any fractionation as noted in Method section title?
8. Does slice thickness affect labelling/yield?
9. Fig S3 – use same colors in a/b.
10. Supp Tables 1 vs 2 – 1552 vs 736 proteins – not clear what the different numbers mean.
11. Supp Table 3/4 – use gene name not prot code/number.

Overall summary of response to review

We thank the Reviewers for their constructive engagement with the manuscript, and identification of areas that needed to be strengthened. The Reviewers considered the study interesting, with proteomics experiments performed to a high standard. The Reviewers' main suggestion was to demonstrate the utility of our genetically targeted proteomics approach either to compare the proteome across multiple cell types or to compare the proteome after a pharmacological or behavioral perturbation. To address this major concern, we have carried out 3 additional new proteomics experiments and extensively edited the manuscript. Below please find a list of updates and new figures.

New proteomics experiments fall into several categories:

1. We performed a high pH reverse phase fractionation of the original samples (comparison between $Drd1^{Cre}$ H2B, NES, LCK APEX) to demonstrate that our APEX method captures sufficient material for deeper proteomic coverage.
2. We compared the baseline proteome between striatal $Drd1^{Cre+}$ direct pathway, dSPNs and $A2a^{Cre+}$ indirect pathway, iSPNs, using the H2B and NES APEX constructs, further demonstrating the cell-type specificity of our approach.
3. We demonstrated that our approach provides sufficient sensitivity to detect cell-type specific proteomic changes after cellular activation. Here, we compared the nuclear and non-nuclear enriched proteome of striatal $Drd1^{Cre+}$ dSPNs with and without cellular activation. We accomplished this by expressing a genetically encoded engineered $G\alpha_q$ -coupled receptor, hM3Dq designer receptor (DREADD). This enabled cell-type and circuit-selective manipulation via an administration of Clozapine N-oxide, its synthetic ligand. This is the first evidence for proteomic landscape modifications by any DREADD, to our knowledge.

Line numbers are referenced throughout the responses. Selected figures are also included in this response for ease of evaluation. The manuscript is greatly strengthened by these additions, providing both technical advances and novel biological insights in the field of neuroproteomics.

Figure	Changes
Figure 1	No change.
Figure 2	Updated schematic in 2a. replaced violin plot and multidimensional clustering with the fractionated dataset.
Figure 3	3b, 3c, 3d, 3e, 3f, 3g, 3f updated with the fractionated dataset. Gene names are italicized. Increased figure and font size.
Figure 4 – New	$Drd1^{Cre}$ dSPNs vs $A2a^{Cre}$ iSPNs comparison
Figure 5 – New	Chemogenetic activation experiments for $Drd1^{Cre+}$ SPNs
Fig S1	No change
Fig S2	No change
Fig S3 – New	Examples of APEX expression in the acute slices and labeling depth experiment
Fig S4 – New	HEK239T subcellular fractionation experiment
Fig S5	(original Fig S3) S5a color changed, S5b, S5c-e updated with fractionation data, S5f,g new panels
Fig S6 – New	Comparison between LCK–NES
Fig S7 – New	Supplemental data associated with Figure 4
Fig S8 – New	Supplemental data associated with Figure 4
Fig S9 – New	Supplemental data associated with Figure 5
Fig S10 – New	Supplemental data associated with Figure 5
Fig S11 – New	Supplemental data associated with Figure 5
Fig S12 – New	NMR for in-house synthesis of propargyl tyramide, a clickable APEX substrate

Supplementary Tables have been re-organized and updated.

Supplementary Table 1. Protein identification and summarization for data in Fig. 2-3 (tab 1: Protein identification without fractionation; tab 2: Protein identification with fractionation; tab 3: Protein quantification by MSstatsTMT for samples with fractionation).

Supplementary Table 2. Statistical comparison for data in Fig. 2-3 (tab 1: APEX.NES vs CTRL; tab 2: H2B.APEX vs CTRL; tab 3: LCK.APEX vs CTRL; tab 4: H2B.APEX vs APEX.NES; tab 5: H2B.APEX vs LCK.APEX; tab 6: LCK.APEX vs APEX.NES; tab 7: a list of H2B-enriched proteins in $\text{Drd1}^{\text{Cre}+}$ dSPNs; tab 8: a list of LCK-enriched proteins in $\text{Drd1}^{\text{Cre}+}$ dSPNs).

Supplementary Table 3. DAVID gene ontology analysis for H2B-enriched nuclear proteins and LCK-enriched membrane proteins for data in Fig. 2-3.

Supplementary Table 4. Protein identification and summarization for data in Fig. 4 (tab 1: Protein identification; tab 2: The contaminant list for filtering out highly abundant non-specific binders and astrocyte-enriched proteins; tab 3: Protein quantification by MSstatsTMT).

Supplementary Table 5. Statistical comparison for data in Fig. 4 (tab 1: H2B.APEX vs APEX.NES for $\text{Drd1}^{\text{Cre}+}$ dSPNs; tab 2: H2B.APEX vs APEX.NES for $\text{A2a}^{\text{Cre}+}$ iSPNs; tab 3: H2B-enriched nuclear proteins; tab 4: non-H2B enriched cytosolic proteins; tab 5: H2B-enriched nuclear comparison (iSPN - dSPN); tab 6: non-H2B enriched cytosolic protein comparison (iSPN - dSPN)).

Supplementary Table 6. DAVID gene ontology analysis for H2B-enriched nuclear proteins for data in Fig. 2-3.

Supplementary Table 7. Protein identification and summarization for data in Fig. 5 (tab 1: Protein identification; tab 2: Protein quantification by MSstatsTMT for nuclear proteins; tab 3: for cytosolic proteins).

Supplementary Table 8. Statistical comparison for data in Fig. 5 (tab 1: APEX.NES vs CTRL; tab 2: H2B.APEX_mCherry vs CTRL; tab 3: H2B.APEX_hM3Dq vs CTRL; tab 4: H2B.APEX_mCherry vs APEX.NES; tab 5: H2B.APEX_hM3Dq vs APEX.NES; tab 6: H2B enriched nuclear protein comparison (hM3Dq – mCherry); tab 7: non-H2B enriched cytosolic protein comparison (hM3Dq – mCherry)).

Supplementary Table 9. DAVID gene ontology analysis for H2B-enriched proteins for data in Fig. 5.

Supplementary Table 10. Differentially regulated transcripts and proteins for the correlation analysis in Fig. 4.

Supplementary Table 11. Custom liquid chromatography gradients for high pH reverse-phase fractionation samples.

Reviewer #1 (Remarks to the Author):

This manuscript from Dumrongprechachan et al describes an interesting proximity labeling approach for cell type- and compartment-specific proteomics in the mouse brain. The authors use a combination of the Drd1-cre mouse line with cre-dependent adeno-associated viruses to express a proximity labeling reagent, APEX, in different subcellular compartments of DRD1+ neurons. Rapid peroxide-dependent labeling was accomplished ex vivo on brain slices. Gel-based analysis confirmed that this labeling system was working as expected. Quantitative TMT proteomics of DRD1+ neuronal proteomes confirmed enrichment of proteins over background, some of which were known (eg Hnrpa2b1) and others that were previously unknown to be enriched in DRD1+ proteomes.

Overall, I found the experiments presented in this manuscript to be interesting and well done. The text is clearly written. The authors have convincingly demonstrated that the APEX system works well in mouse brain. However, my enthusiasm for the manuscript in its current form is still limited. This is largely because there is already similar published work on cell type-selective proteomics or proximity labeling in mouse brain (PMID 29251727, 29106408, 27609886, 30674877, 33066078). This manuscript would be a major advance and distinct from this existing literature if they could demonstrate either generality across multiple cell types or applicability for dynamically changing proteomes. While these applications are provocatively suggested in the text, no experimental data are provided for their feasibility. If the authors could address this issue, in my opinion this work would be of broad interest to the community and wider field.

We thank the Reviewer for describing our work as interesting and well done. We have extensively revised our manuscript and added all experiments recommended by the Reviewer. We have addressed and demonstrated applications of the APEX strategy in the mouse brain across cell types (new **Fig. 4**) and for dynamically changing proteomes (new **Fig. 5**).

New Figure 4. APEX-based comparison between the nuclear and cytosolic proteomes of dSPNs and iSPNs.

- (a) Targeted expression of APEX in direct and indirect-pathway spiny projection neurons (dSPNs and iSPNs).
- (b) Cre-dependent APEX-NES expression in the striatum of A2a^{Cre} mouse line. *Left*, sagittal section (1mm) of A2a^{Cre} > 2 weeks post AAV transduction. *Middle*, somatic fill EGFP reporter in the striatum (STR) (100 μ m). *Right*, iSPN-specific axonal projections in the external globus pallidus (GPe) (200 μ m).
- (c) Western blot analysis of biotinylated striatal lysates prepared from Drd1^{Cre} and A2a^{Cre} mouse lines.
- (d) Subcellular localization of APEX constructs in striatal iSPNs. Confocal images of APEX-expressing neurons including EGFP, immunostained FLAG-Ab, and streptavidin for visualizing APEX subcellular localization and biotinylation patterns (20 μ m).
- (e) Statistical analysis workflow for comparing dSPN and iSPN proteomes.
- (f) Gene ontology analysis of the 458 H2B-enriched nuclear proteome generated from H2B – NES comparison. Non-H2B enriched proteins are considered as a part of the cytosolic proteome.
- (g) Differential expression analysis for the non-H2B enriched cytosolic proteome (A2a^{Cre}.NES – Drd1^{Cre}.NES).
- (h) Differential expression analysis for the H2B-enriched nuclear proteome (A2a^{Cre}.H2B – Drd1^{Cre}.H2B).
- (i) Workflow for correlation analysis between proteins and transcripts.
- (j) Correlation between log₂FC (dSPN – iSPN) between proteins and translating mRNA TRAPseq dataset
- (k) Same as (j) for mRNA expression from single-cell RNAseq dataset.

New Figure 5.

Differential expression analysis of the dSPN proteome following CNO-induced activation of hM3Dq.

- (a) Experimental paradigm. Targeted co-expression of APEX and hM3Dq or fluorophore control in *Drd1^{Cre+}* dSPNs. Clozapine N-oxide (CNO) was administered (3 mg/kg i.p.) to selectively activate dSPNs via $G\alpha_q$ -coupled signaling cascades. Following CNO treatment, acute slice preparation and *ex vivo* biotinylation were performed to measure proteomic changes at 2.5–3 hr post CNO administration.
- (b) Western blot analysis of biotinylated striatal lysates for samples used in this dataset.
- (c) Statistical analysis workflow for comparing the effect of CNO-induced changes in the dSPN proteome between hM3Dq-expressing and fluorophore control samples.
- (d) Hierarchical clustering heatmap of proteins and biological replicates.
- (e) Gene ontology analysis of H2B-enriched and NES-enriched protein lists.
- (f) Scatter plot of \log_2FC (mCherry-NES) v.s. \log_2FC (hM3Dq-NES) for the H2B-enriched nuclear proteome. Blue and red dots are downregulated and upregulated proteins (q -value < 0.05).
- (g) Differential expression analysis for the nuclear proteome (hM3Dq – mCherry) (dotted line: q -value = 0.05).
- (h) Differential expression analysis for the cytosolic proteome (hM3Dq – mCherry) (dotted line: q -value = 0.05).
- (i) Immunostaining of *JunB* in the striatum 2.5 hrs post CNO administration (20 μ m).
- (j) Reporter ion quantification of JunB interacting proteins that were identified only by one unique peptide.
- (k) Cytoscape network of JunB protein interactions detected in the dataset (color: \log_2FC (hM3Dq – mCherry)).

Major comments:

1. As mentioned above, this manuscript would be greatly strengthened if the authors could demonstrate the generality of their method. Obviously many different cre drivers are available for genetic manipulation in the brain. It would be of great interest to compare nuclear proteomes across a few different types of neurons (or even between neurons and non-neuronal cell types such as glia).

Here, we performed an additional experiment to compare the nuclear proteome of direct- and indirect pathway spiny projection neurons (dSPNs and iSPNs) by expressing H2B.APEX and APEX.NES in the striatum of *Drd1^{Cre}* and *A2a^{Cre}* transgenic lines, respectively (new Fig. 4). The classical markers of the two SPNs, *Drd1* and *A2a* receptors, were properly classified for dSPNs and iSPNs, respectively. Moderate correlation between transcripts and proteins ($r=0.61$, 0.46 for TRAPseq and scRNAseq datasets) was observed (new Fig. 4i-k) in line with many other investigations of the congruence between transcript and protein levels. In addition to DA and adenosine receptors, we identified several other differentially expressed nuclear and cytosolic proteins (new Fig. 4g, h) (new Supplementary table 5). Overall, we demonstrate the generality of our approach for mapping compartment and cell type specific proteomes in mouse brain. Lines: 235–286.

2. Another major advantage of this method could be to monitor proteome dynamics on short time scales, an application that is not easily achieved by unnatural amino acid incorporation methods due to slow metabolic labeling. What happens to *Drd1* proteomes following pharmacological or behavioral challenge?

To demonstrate that the APEX strategy provides sufficient depth, accuracy, and precision to examine activity-dependent changes in the proteome in a single cell class, we selectively activated dSPNs using a widely used chemogenetic hM3Dq designer receptor (new Fig. 5). By co-expressing hM3Dq and H2B.APEX in dSPNs, we captured changes that occur in the first 2.5–3 hr after neuronal activation by hM3Dq ligand, clozapine N-oxide. Key changes in the nuclear proteome of dSPNs include the up-regulation of the immediate early genes (IEGs) such as *Junb*, *Egr4*, *Nr4a1*, etc. Overall, we show that our *ex vivo* preparation efficiently captures changes induced by *in vivo* manipulation, validating the APEX workflow for activity-dependent investigations of proteome dynamics with cell-type and compartment specificity. Lines: 287–353.

Minor comments:

3. It would be helpful if the authors could validate at least some of their proteomic data by staining or some other orthogonal approach. Are these identified proteins actually enriched in DRD1+ neurons, or are they general nuclear neuronal markers?

First, following the validation of APEX constructs in *Drd1^{Cre}* dSPNs (updated Fig. 2-3), we additionally compare the nuclear proteome of dSPNs and iSPNs (new Fig. 4). The latter dataset shows that majority of *Drd1^{Cre}* nuclear proteins detected in the prior validation dataset (updated Fig. 3) are general nuclear markers for both SPNs, measured by *A2a^{Cre}* – *Drd1^{Cre}* comparison (e.g., *Hnrnpa2b1*, log₂FC = -0.043, p = 0.607, and *Hnrnpd*, log₂FC = -0.011, p = 0.89). Lines: 278–283.

Second, for the chemogenetic experiment in new Fig. 5, we identified *Junb* with the greatest log₂FC in our dataset from the MSstatsTMT pipeline. Therefore, we repeated the chemogenetic experiment and performed immunofluorescence staining for further validation. We confirmed that *Junb* is upregulated in hM3Dq-expressing dSPNs, consistent with the proteomic data (new Fig. 5i). Lines: 335–338.

Updated Figure 2. APEX2-based cell-type specific proteomics in the mouse striatum.

- Workflow for proteomics sample preparation. Following *ex vivo* acute slice biotinylation, biotinylated proteins were enriched with streptavidin beads. Beads were washed and digested to generate tryptic peptides. Peptides were TMT labeled and mixed equally before high pH reverse-phase fractionation. MS data were acquired using the synchronous precursor selection (SPS) MS³ method with real-time search (RTS) against the SwissProt database. Peptides were identified by Proteome Discoverer at 1% false discovery rate (FDR) and summarized by MSstatsTMT.
- Western blot analysis of biotinylated brain lysate and flowthrough fractions. Equal amounts of proteins and streptavidin beads are used in the enrichment process. Depletion of biotinylated proteins in the flowthrough fractions confirms successful enrichment.
- Western blot analysis of eluate fractions after streptavidin bead enrichment. Differential enrichment output reflects varying amounts of biotinylated proteins from each sample.
- Overall protein intensities across all samples.
- Multidimensional scaling plot approximates expression differences between the samples for the top 500 proteins.

Updated Figure 3.

In situ proximity labeling generates cell-type and subcellular compartment specific proteomes.

- Statistical analysis workflow. Moderated t-test for significance with Benjamini-Hochberg (BH) multiple hypothesis correction is used as a cut-off. Filter 1 is the first comparison to determine the degree of protein enrichment above non-specific background. Cre-positive samples are compared against Cre-negative control to remove enrichment contaminants. Filter 2 is used with proteins that were retained after the first filter to determine the extent of subcellular enrichment.
- Differential enrichment analysis for *Drd1*^{Cre} dSPN (H2B – CTRL). Proteins significantly enriched above background are indicated by red dots ($q\text{-value} < 0.05$ and $\log_2\text{FC} > 0$).
- Differential enrichment analysis for *Drd1*^{Cre} dSPN (H2B – NES) and (H2B – LCK). *Left*, nuclear enriched proteins (H2B) are indicated by red dots. *Right*, nuclear enriched proteins (H2B), red dots; membrane-enriched proteins (LCK), blue dots.
- Gene ontology cellular component (GO CC) analysis for H2B-enriched and LCK-enriched proteins.
- Log₂FC vs rank plot for H2B-enriched nuclear proteins. Proteins with UNIPROT nucleus annotation, red dots, a subset of proteins labelled. Positive log₂ fold change (H2B-NES) indicates biased enrichment towards the nucleus.
- Log₂FC vs rank plot for LCK-enriched membrane proteins. Proteins with UNIPROT membrane annotation, blue dots, a subset of proteins labelled. Negative log₂ fold change (H2B-LCK) indicates biased enrichment towards the membrane.

Reviewer #2 (Remarks to the Author):

The manuscript delivers on providing a demonstration of the ability to enrich and identify proteins from Drd1 positive cells and enrich for proteins from different compartments. This approach will be of interest to the neuroscience community and will enable a better understanding of the proteins involved in achieving different functions in brain circuitry. The value of the subcellular compartments is less well justified by the authors. They appear to be the first to demonstrate the overall approach. The proteomics and statistics methodology were appropriately performed and to a high standard.

We thank the Reviewer for deeply engaging in our work and describing our proteomics and statistics methodology performed and to a high standard. We conducted multiple new experiments, as well as improving the main text and the method sections, as recommended by the Reviewer.

There is a major issue with the description of the control for the proteomics analysis. The control is not described anywhere, except in the text as “Cre-negative”. One presumes that C57BL/6 were used. However, the only mention of C57BL/6 mice is: “C57BL/6 mice used for breeding and backcrossing”. It is also possible that a mock injection was done, or a vector containing EGFP, either with or without the DIO, was used as control on Drd1(Cre) mice. The latter may have been a better control than using C57BL/6 mice. Nothing was written to help the reader. A control (AAV8.hSyn.DIO.Cherry) is used for electrophysiology. There’s no indication whether that control was also used in the proteomics; presumably not because it would be strange to call it “Cre-negative”.

Drd1^{Cre}(EY262) are hemizygote animals (including the A2a^{Cre} used in the new experiment new **Fig. 4**, page 4 of response to review). Therefore, littermates are a mixture of both Cre-positive and Cre-negative animals. During the stereotactic injection, we injected the entire litter and genotyped for the presence of Cre transgenes after AAV injection. In other words, the Cre-negative control group in the proteomics experiment (updated **Fig. 2–3**, pages 8 and 9 of response to review) were mice that received AAV injection neonatally (one of the three DIO.APEX viruses), but they did not express APEX. In addition, tissues prepared from Cre-negative littermate mice underwent the same and concurrent biotinylation and enrichment procedure, serving as the optimal negative enrichment control. In contrast, for electrophysiology, we used Cre-positive animals with a fluorophore AAV control, in order to visualize neurons for recording.

We have included the following statements in the revised manuscript’s method section for better clarification of the control subjects for proteomics experiments. **Lines:** 626–629.

“P35-P70 animals were used for histology, western blots, and proteomics experiments. Drd1^{Cre}-negative littermates, which also received one of the three APEX AAVs, were used as a negative enrichment control group for proteomics, because Drd1^{Cre}-negative littermates do not express DIO.APEX.”

The control is important because it gets to the heart of whether this data is biologically useful. We are presented with: (a) lists of proteins enriched in Drd1+ cells and (b) lists of proteins enriched in the nucleus and membrane of Drd1+ cells. The list of Drd1+ cell enriched proteins is not particularly useful when compared to the control. The control is mainly a list of proteins that bind the beads non-specifically plus those mitochondrial proteins which are endogenously biotinylated. The list does not tell us whether these proteins are enriched in this cell type compared to any other brain cell type. Thus, the value is limited. The only immediate way to increase value would be to validate one/some of these results. Then we have the list of compartment-enriched proteins. The list helps identify some proteins, “Dock3 and Actn4”, that may be newly identified as membrane proteins. However, the immediate use of defining compartments is not obvious. Furthermore, the membrane and cytosolic compartments had no significant differences. Perhaps if the proteins changed compartment following stimulation, then there will be mechanistic value. When considering the wording of the abstract, introduction and concluding sentences, the article is sold on the value of the cell type-specific analysis, but the main deliverable is the nucleus and membrane compartment comparison.

We agree with the Reviewer that the pull-down negative control in this experiment is critical to evaluating whether proteins are enriched from Cre-positive cells (part a, from Reviewer comment). Then, the next comparison between APEX constructs generates lists for the nucleus and the membrane proteome (part b, from Reviewer comment) (updated **Fig. 3**, page 9 of response to review). Although these lists do not tell us whether these proteins are enriched in this cell type compared to any other brain cell type, they inform us on the relative protein abundance in a particular cell class (e.g., Drd1^{Cre+} neuronal proteome over striatal tissue background). This is an important validation step to determine that our affinity enrichment allows us to profile the proteome of one cell type.

We conducted new proteomics experiments: (i) to compare the proteome between striatal Drd1^{Cre+} and A2a^{Cre+} striatal spiny projection neurons (new **Fig. 4**, page 4 of response to review), and (ii) to identify changes in the proteome of Drd1^{Cre} dSPNs after *in vivo* neuronal activation (new **Fig. 5**, page 6 of response to review) (see also Reviewer #1 major comment #1 and 2, page 7 of response to review). We note here that compartment resolution in these experiments enhanced our ability to

detect differences between cell types and with/without neuronal activation. **Lines:** 235–286 for **Fig. 4** and **Lines:** 287–353 for **Fig. 5**.

The dataset generated in new **Fig. 4** (page 4 of response to review) is one of the few cell-type specific proteomic maps of dSPNs, and iSPNs with the proteome coverage spanning the somatodendritic and nuclear compartments. This experiment provides the protein-level evidence for many differentially expressed transcripts. The current datasets provide numerous targets for directed follow up investigations to determine the roles of differentially expressed proteins in regulating SPN physiology. Altogether, we demonstrate the utility of our approach for broad neuroscience applications.

When considering that the depth of the proteome analysis is not particularly deep (1,552 proteins. 736 quantified. They concede that fractionation prior to LC-MSMS could have helped), the overall impression is that this is a new approach that has been rushed to publication as an article of high novelty, but low immediate biological value, especially to those with interest in Drd1+ cells. The article may have immediate value to those interested in performing a similar cell type-specific analysis. Overall, I am concerned that the manuscript has low value beyond its novel approach.

We have performed high pH reverse phase fractionation of the original samples for a new side by side comparison, to precisely quantify the gains associated with fractionation. The depth of the proteomic analysis has increased, reported in new panels **Supplementary Fig. 5f, g**. Data in the **Fig. 2-3** (pages 8–9 of response to review) were updated with fractionation data, with IDs increasing from 1570 to 4365 and quantification from 656 to 2191. **Lines:** 176–179.

Updated Supplementary Figure 5. Analysis of MS sample preparation reproducibility.

- Experimental design schematic. Samples were randomized across two TMT 10-plex sets (top and bottom). Direction left-to-right was ordered by TMT channels (126-131). Each set contained 2 complete blocks (each block = Norm, NES, H2B, LCK, Cre-).
- Multidimensional clustering of log₂ protein intensities without reference ID channel normalization. Each point was a biological replicate. Arrow indicates a clear separation between TMT sets, when reference normalization was not performed.
- Multiscatter plot of H2B biological replicates (example). *Diagonal*, histograms of protein log₂ intensities. *Upper right corner*, Pearson correlation coefficient r , *Lower left corner*, pairwise scatter plots between replicates, red lines are loess-fit lines.
- Box plot of %CV distribution.
- Density plot of %CV distribution.
- Protein identification and quantification before and after high pH-reverse phase fractionation.
- Overlap between protein ID before and after high pH-reverse phase fractionation.

Minor issues:

P2A not explained as self-cleaving element of their vector.

We have made changes to the following text. **Lines:** 101–105.

“We demonstrated Cre-dependent expression by neonatal viral transduction in the striatum of *Drd1^{Cre}* mice with adeno-associated viral vectors (AAVs) encoding one of three APEX variants and a EGFP reporter with a ribosomal skipping P2A linker (**Fig. 1a**). Because P2A-linked EGFP is translated separately from APEX, EGFP is expected to distribute throughout the cell, while APEX is directed to targeted subcellular domains”

“SNr” abbreviation not defined.

We have added the full name of SNr to the following text. **Lines:** 106–107.

“We verified Cre-dependent expression of our AAVs by examining the projection pattern of the EGFP reporter. *Drd1^{Cre+}* dSPNs send axonal projections primarily to the substantia nigra pars reticulata (SNr). In all APEX variants (**Fig. 1b** and **Supplementary Fig. 1a**) EGFP expression was similar across constructs, as expected for P2A-linked EGFP, where the distribution is independent from the APEX localization and should be the same regardless of APEX targeting.”

“SPS MS3”. “SPS” not defined.

We have defined the “SPS MS3” abbreviation in the following text. **Lines:** 169–172.

“Peptide sequences were identified using the MS2 spectra, while the quantification of reporter ions was performed using the synchronous precursor selection (SPS) MS3 method²⁷ with the real-time search (RTS)²⁸. During data acquisition, RTS triggers MS3 reporter quantification when MS2 fragmented spectra are matched to the mouse SwissProt database, improving MS3 quantification efficiency and accuracy²⁸.”

Beginning of 10th paragraph “Genetically targeted proximity labeling approaches offer information on cell-type specificity and subcellular localization.” The sentence seems repetitive and unnecessary here. Suggest deleting.

This sentence has been removed in our revised manuscript.

A minor, but troublesome issue is whether the streptavidin binding sites on the beads were in excess compared to the biotinylated beads. If not, the biotinylated proteins, including endogenous biotinylated proteins, would be in competition for streptavidin binding sites and this may affect any comparisons if, for example, there were more endogenous biotinylated proteins in one sample than another. Ideally, an experiment would be performed to show at what point a fixed amount of beads is able to extract maximum protein from a varying level of lysate (and then a lysate level is used below this saturation point to ensure minimal competition).

We thank the Reviewer for raising this concern. We agree that the competitive nature of streptavidin-biotin enrichment can be problematic. Although we have not established the amount of beads per lysate level to find the minimal competition point, we are able to identify many true positives for given APEX construct, cell class, and activity states using our statistical analysis approach. True positives are, for example, (i) nuclear proteins that are correctly classified using the H2B.APEX construct, (ii) classical molecular marker for SPN subtypes (e.g., *Drd1* for dSPNs, and *A2a* for iSPNs), and (iii) immediate early genes such as *JunB*, *Nr4a1*, *Egr4*, that are upregulated after neuronal activation.

To reduce technical variability stemmed from streptavidin enrichment, we used the same fixed amount of beads per sample within each experiment and heavily rely on the overall reproducibility during sample preparation. Key to our success is the criteria to exclude subjects with suboptimal APEX expression (which would lead to poor biotinylation level). To demonstrate that the reproducibility of the biotinylation workflow, we evaluated biotinylation level in cell lysates using western blot prior to enrichment (new **Supplementary Fig. 7b** for SPN comparison, and new **Supplementary Fig. 10b** for the chemogenetic experiment). In addition, we have provided detailed protocol including the amount and volume of protein, beads, and reagents in the method section. **Lines:** 763–827.

New Supplementary Figure 7. Characterization of APEX expression in striatal A2a^{Cre+} neurons and experimental design for SPN comparison.

- (a) Cre-dependent expression patterns of APEX variants in the striatum and the GPe of the A2a^{Cre} mouse line. EGFP, DAPI, immunostained FLAG-Ab for APEX were in green, blue, and red, respectively. Scale bar: 0.5 mm
- (b) Western blot analysis of biotinylated lysates before MS sample preparation. Dotted line shows cropped region for blot in Fig 4b.
- (c) Relative peptide output after on-bead digestion before TMT labeling.
- (d) TMTpro 16plex experimental design.

New Supplementary Figure 10.
Experimental design and sample quality check for the chemogenetic experiment.

- (a) TMT 11plex experimental design.
- (b) Western blot analysis of biotinylated lysates before MS sample preparation. Dotted line shows cropped region for blot in Fig bc.
- (c) Relative peptide output after on-bead digestion before TMT labeling.

“This was not surprising because mitochondrial compartment is oxidative.” This reason is a little speculative and should be further supported if the authors insist on using it. Mitochondrial contamination could simply be due to the high abundance of mitochondrial proteins or indirect binding of mitochondrial proteins to biotinylated mitochondrial proteins (i.e. extraction of mitochondrial protein complexes).

We thank the Reviewer for carefully evaluating our work. We have removed the statement from the revised manuscript.

“Conditional expression of APEX and reporters in Cre+ neurons was achieved by recombinant adeno-associated viral transduction encoding a double-floxed inverted open reading frame (DIO) of target genes, as described previously.” There should be a reference at the end of this sentence. “double-floxed inverted open reading frame (DIO)” abbreviation appears above this in “Plasmid construction and AAV preparation” section.

We have addressed this concern through the following edits:

- (i) We defined DIO in the earlier “Plasmid construction and AAV preparation” section. **Lines:** 599–601.

“APEX2-NES-P2A-EGFP, LCK-APEX2-P2A-EGFP, and H2B-APEX2-P2A-EGFP were synthesized by Genscript and subcloned into pAAV-EF1a-DIO-WPRE-hGH vector (a gift from Dr. Karl Deisseroth, Addgene plasmid #20297) using restriction enzymes Ascl and NheI. DIO abbreviation refers to the double-floxed inverted open reading frame.”

- (ii) We have provided appropriate citations for neonatal transduction that were missing in the original manuscript. **Lines:** 615–617.

“Conditional expression of APEX, hM3Dq, and reporters in Cre+ neurons was achieved by recombinant adeno-associated viral neonatal transduction encoding a double-floxed inverted open reading frame (DIO) of target genes, as described previously^{49,50}.”

It should be stated in the methods that the minimum number of peptides required for protein identification was 1. This is reported in the table but must be stated in the methods.

We added additional clarifications in the method section:

1. For protein identification. **Lines:** 887–889.

“PSMs were filtered by the Percolator node (max Delta Cn = 0.05, target FDR (strict) = 0.01, and target FDR (relaxed) = 0.05). Proteins were identified with a minimum of 1 unique peptide and protein-level combined q values < 0.05.”

2. For protein quantification. **Lines:** 894–901.

“PSMs was exported from Proteome Discoverer and converted into MSstatsTMT-compatible format using PDtoMSstatsTMTFormat function. PSMs were filtered with a co-isolation threshold = 70 and peptide percolator q value < 0.01. For protein quantifications, only unique peptides were used. In addition, only proteins with a minimum of two unique PSMs were quantified (e.g., proteins with 1 unique peptide must have at least two PSMs of different charges to be considered for summarization in the MSstatsTMT). Therefore, not all identified proteins were quantified. Protein summarization was performed in MSstatsTMT with the following arguments: method = msstats, global median normalization = FALSE, reference normalization = TRUE, imputation = FALSE.”

Having text that is 2-4 point in size, as in Fig. 3e, is not desirable. I suggest making the figure larger. “Dock3 and Actn4” are gene names. When referring to the proteins, protein names should be used. Of course, genes names can be used, but it should be obvious that they are not the same as the protein names. Mouse gene names should be italicised, including when shown in figures (e.g. Fig. 3e).

We have italicized all gene names used in the text and figures.

Reviewer #3 (Remarks to the Author):

This manuscript describes a proteomics method to identify cell type and subcellular compartment specific proteomes in the mouse brain by using genetically targeted proximity biotinylation labeling employing APEX2. Specific subcellular compartments were interrogated by creating three Cre-dependent APEX variants that localized to the nucleus, to the cytosol, and to the membrane compartment by tagging with well validated sorting motifs (Histone 2B fusion, H2B; nuclear exporting sequence, NES; and membrane anchor LCK sequence, respectively) and coupling with TMT analysis. A cell type-specific component was included by use of expression of the APEX2 variants in *Drd1* striatal medium spiny neurons. Overall, the authors were able to identify 1552 proteins (but see comment below) and successfully classified differential localization of 10s of proteins specific to cell type and subcellular compartment.

Overall, this is a solid study, with careful design and analysis. The manuscript is well written and the interpretation of the results is appropriate. The work expands examples of the utility of APEX2 in proximity biotinylation studies. However, the study is missing an essential element, namely a comparison of 2 or more different cell types. An obvious comparison would be with *Drd2*-expressing medium spiny neurons. Alternatively, comparison with cerebellar, cortical or hippocampal neurons would be informative. Only then would one know if the approach has sufficient proteomic depth to identify and quantify differentially expressed gene products, and to be comparable to scRNAseq or TRAP approaches. There is also little new biological insight provided. Some sort of drug treatment or behavioral perturbation would be needed.

We thank the Reviewer for describing our work as a solid study with careful design and analysis. We have addressed all the Reviewer's specific comments as well as adding new proteomics datasets to strengthen our study as suggested by the Reviewer, including (i) a comparison between *Drd1* and *A2a*-expressing striatal spiny projection neurons in new **Fig. 4** (page 4 of response to review), and (ii) a comparison of *Drd1* dSPNs after neuronal activation in new **Fig. 5** (page 6 of response to review). Altogether, new proteomic experiments demonstrate the application of our APEX workflow to compare the proteome between two neuronal types, and across activity states of neurons. **Lines:** 235–286 for **Fig. 4** and **Lines:** 287–353 for **Fig. 5**.

Specific comments:

1. The authors show the subcellular specific localization of Cre-dependent APEX variants by microscopy. Did the authors also examine this by western blot after fractionation of nucleus, cytosol and membrane fraction from total striatal tissue.

We performed subcellular fractionation to enrich proteins from cytosolic, membrane, and nuclear compartments from HEK293T cells expressing one of the three APEX constructs (H2B, NES, and LCK) in new **Supplementary Fig. 4b**. We used western blot analysis to assess the biotinylation pattern. **Lines:** 149–154.

The total protein staining loading control shows that the nuclear, cytosolic, and membrane fractions can be qualitatively distinguished (denoted by N, C, M lanes, respectively). As for biotinylation patterns, H2B.APEX shows the most distinct pattern mimicking the total protein staining of the nuclear fraction (**lane 4**). In contrast, biotinylation patterns between LCK and NES constructs in HEK293T are similar (**lanes 7-12**). Our interpretation is that in HEK293T cells, biotin-phenol generated by the LCK.APEX and APEX.NES are diffusible and accessible to similar sets of the proteome (*i.e.*, proteins that are shared across the cytosolic and membrane compartments).

New Supplementary Figure 4b. Western blot analysis of APEX biotinylation patterns across different subcellular fractions in HEK293T. N: nuclear fraction, C: cytosolic fraction, and M: membrane fraction.

2. The authors should provide separate lists of identified proteins in each compartment.

A separate list of proteins differentially enriched in each compartment is included in new **Supplementary table 2 (tabs 7 and 8)**. Similarly, differentially enriched proteins in the nuclear compartment in new experiments (new **Fig. 4 and 5**) are provided in new **Supplementary table 5 (tabs 3 and 4)** and new **Supplementary table 8 (tabs 6 and 7)**, respectively.

3. The authors should provide more details about data processing like how many minimum peptides were used for protein IDs and peptide and protein FDR in the methods section.

We added additional clarifications in the method section:

1. For protein identification. **Lines:** 887–889.

“PSMs were filtered by the Percolator node (max Delta Cn = 0.05, target FDR (strict) = 0.01, and target FDR (relaxed) = 0.05). Proteins were identified with a minimum of 1 unique peptide and protein-level combined q values < 0.05.”

2. For protein quantification. **Lines:** 894–901.

“PSMs was exported from Proteome Discoverer and converted into MSstatsTMT-compatible format using PDtoMSstatsTMTFormat function. PSMs were filtered with a co-isolation threshold = 70 and peptide percolator q value < 0.01. For protein quantifications, only unique peptides were used. In addition, only proteins with a minimum of two unique PSMs were quantified (e.g., proteins with 1 unique peptide must have at least two PSMs of different charges to be considered for summarization in the MSstatsTMT). Therefore, not all identified proteins were quantified. Protein summarization was performed in MSstatsTMT with the following arguments: method = msstats, global median normalization = FALSE, reference normalization = TRUE, imputation = FALSE.”

4. Two filter criteria were used to isolate cell type and subcellular specific proteomes 1) comparisons between Cre-positive samples and Cre-negative control samples; 2) stringent cutoff for the nuclear and membrane-enriched proteomes, using moderate *t*-tests. Can the authors explain about handling the proteins which are present in two places like nucleus and cytosol. Maybe these proteins are filtered out during differential analysis. This might explain not getting any different proteins in the NES-LCK comparison.

We agree with the Reviewer that many proteins localize to multiple compartments. In new **Supplementary Fig. 6b**, we show that majority of LCK+ proteins are shared between membrane and cytosol. We also observed modest log2FC between LCK and NES (approximately $-1 < \log_2FC < 1$, new **Supplementary Fig. 6c**), compared with the H2B vs NES comparison (approximately $-2 < \log_2FC < 3$, updated **Fig. 3c**). This suggests that LCK.APEX and APEX.NES can label similar proteins. This could explain why protein entries in LCK-NES comparison did not pass the stringent statistical cutoff ($q < 0.05$). Alternatively, when a less stringent cutoff is used (e.g., nominal p-value < 0.01), the LCK construct is shown to preferentially label the membrane compartment over the cytosol (new **Supplementary Fig. 6c**), with many membrane-associated proteins annotated in the volcano plot (e.g., *Atp2b1*, *Gabrb3*, *Ncam2*, *Gria3*).

New Supplementary Figure 6. Biotinylation by APEX.NES and LCK.APEX.

- Comparison between APEX.NES or LCK.APEX and Cre-negative control.
- Subcellular annotation for LCK-CTRL enriched proteins. Venn diagram showed that majority of LCK-CTRL enriched proteins have both membrane and cytoplasm/cytosol annotations.
- Comparison between LCK.APEX and APEX.NES.
- Log₂ ratio vs rank plot for LCK – NES. Greater log₂ (LCK – NES) ratios indicated a greater enrichment by the LCK construct.

5. Fig 1F – why does the protein stain look identical in all conditions? Are these abundant endogenously biotinylated proteins found in all samples.

In previous and current Fig. 1f, we used total protein lysates for western blot analysis to confirm that biotinylation in the acute brain slices depends on APEX, biotin-phenol, and H₂O₂. This was done prior to streptavidin bead enrichment, therefore, total protein should be approximately equal.

Total protein stain (Fig 1f, right) serves as a loading control to show that omission of one reaction components did not lead to significant biotinylation (Fig. 1f, left). In streptavidin channel, the doublet bands (~125 kDa and ~70 kDa) corresponds to putative endogenously biotinylated proteins which were found in all samples (putative ID: 129 kDa pyruvate carboxylase, and 79 kDa mitochondrial propionyl-CoA carboxylase alpha chain). Both endogenous biotinylated proteins (UniprotID: Q05920 and Q91ZA3, respectively) were detected and filtered out in our proteomics data as enrichment contaminants.

Previous and current Figure 1f.

Conditional protein biotinylation in acute brain slices. APEX-mediated biotinylation requires BP and H₂O₂.

6. *Ex vivo slice dissection* – how consistent was the fluorescent signal through a section/slice?

In a separate injection cohort from the proteomics dataset, we prepared acute brain slices to examine injection variability. We included examples of no APEX control (Cre-negative animals that received one of the AAV.DIO.APEX injection) and APEX-EGFP+ acute slices in new **Supplementary Fig. 3a-b**. We included example images from injections that would be included for the proteomics analyses as well as excluded, based on low expression. **Lines:** 134–136.

Tissue background fluorescence was observed when we imaged the Cre-negative sections at 100 ms exposure time, while APEX-EGFP+ expression in Drd1^{Cre+} slices is evidently detected at 10 ms exposure time. There was some injection (*i.e.*, expression) variability across three example animals (APEX 1–3 samples). For example, subject APEX 3* shows lower EGFP signal to background (with enhanced brightness adjustment compared to APEX 1 and 2). The signal in the somata (top arrow) was dim, almost matching that of projection tract (bottom arrow). Animals with suboptimal expression due to a mistargeted injection were evaluated during the acute brain slice preparation and were not used for proteomics analysis.

New Supplementary Figure 3a-b.

- Examples of APEX expression variability in the acute slices. Each section represents an individual subject. Cre-negative control animals received AAV transduction but did not express APEX. Background tissue fluorescence signal was detected at 100 ms exposure.
- Same as (a) for Drd1^{Cre+}. Expression of APEX-EGFP+ in the dorsal striatum was detected at 10 ms exposure. * denotes a subject with suboptimal expression shown with enhanced brightness adjustment relative to the first two (lower EGFP signal to tissue background).

7. TMT methods – was there any fractionation as noted in Method section title?

In the original manuscript, there was no fractionation. In this revised manuscript, we have performed high pH reverse phase fractionation of the original samples, for a side-by-side comparison of proteome coverage. **Lines:** 174–179. The method section has been updated to include fractionation protocol. **Lines:** 805–827.

“The remaining samples (12 μ l) were mixed equally and dried in a vacuum concentrator for high pH reverse-phase fractionation (Cat. No. 84868, Thermo Fisher). Briefly, samples were resuspended and sonicated for 10 min in 300 μ l buffer A (LC MS water 0.1% FA) supplemented with 1 μ l FA. Sample acidity was verified by pH papers. Resin was packed by centrifugation at 5,000 g for 2 min, activated twice with 300 μ l ACN, and conditioned twice with 300 μ l buffer A. Peptides were loaded five times by centrifugation at 3,000g for 2 min. Column was first washed with 300 μ l water, and eluted by increasing percentage of acetonitrile in 0.1% triethylamine solution according to the manufacturer instructions (5%, 10%, 12.5%, 15%, 17.5%, 20%, 22.5%, 25%, and 50% ACN). All fractions were dried in a vacuum concentrator.”

The updated **Fig. 2-3** are now updated with fractionation data (pages 8–9 of response to review). Depth of protein identification and quantification can be found in new **Supplementary Fig. 5f-g** (page 12 of response to review). In addition, we carried subsequent experiments with fractionation (new **Fig. 4** and **5**) (pages 4, 6 of response to review).

8. Does slice thickness affect labelling/yield?

A typical thickness for acute brain slices ranges from 250 μ m to 300 μ m to preserve local circuit connectivity. We added a new experiment in new **Supplementary Fig. 3c-d** to evaluate labeling depth. We found that APEX labeling is depth-dependent and occurs at least \sim 70 μ m deep, as far as our detection reagent can reach. This experiment places a lower bound on the estimated labeling depth. **Lines:** 135–147.

Supplementary Figure 3c-d.

- Workflow for APEX labeling depth analysis in the acute slices using scanning oblique plane illumination light-sheet microscopy. Acute slices were incubated in ACSF with propargyl tyramide (PT) for 1 hr. Slices were fixed with 4% PFA prior to click reaction for 1-2 hr.
- Cross section of acute slices imaged. PT-labeled neurons were detected as deep as \sim 70 μ m below the surface.

9. Fig S3 – use same colors in a/b.

In the updated manuscript, Fig S3 is re-labeled as updated **Supplementary Fig. 5a-b** (page 12 of response to review). The same colors are now used.

10. Supp Tables 1 vs 2 – 1552 vs 736 proteins – not clear what the different numbers mean.

The original numbers are the number of protein identification by Proteome Discover with a minimum number of unique peptides = 1, and the number of proteins quantified by the MSstatsTMT package, respectively. We have clarified the differences between protein identification and quantification in the method sections. **Lines:** 887–889, and 894–901.

Please see also Reviewer#3 – specific comment #3. We have updated the number of protein IDs and quantification throughout the figures (**Supplementary Fig. 5f, Fig. 4e, and Fig. 5c**) (page 12, 4, 6 of response to review, respectively).

11. Supp Table 3/4 – use gene name not prot code/number.

We have updated relevant tables to include both UniProt accession and gene names. The original supplementary table 3 and 4 are now summarized into the new **Supplementary table 2, tabs 1-6**.

REVIEWER COMMENTS

Reviewer #1 (Remarks to the Author):

The authors have satisfactorily addressed all of my previous questions. I appreciate the two entirely new experimental figures that directly answer questions regarding generality of this approach across cell types and the dynamic modulation of the proteome in response to chemogenetic activation.

Reviewer #2 (Remarks to the Author):

Overall the manuscript is greatly improved by the addition of a cell type comparison and a dynamic change due to stimulation. The manuscript will now be of high interest to the community, although mainly for its methodological aspects and cell type comparison. The dynamic protein changes don't make any new discoveries. The work is done to a high standard with good methodology and statistics, including the additions. Some minor outstanding issues are below.

I mentioned previously that competition from endogenously biotinylated proteins could interfere with protein extraction, particularly when the streptavidin is not in excess. The authors did not test this. The issue could affect the ranking of proteins in their data. I don't think this should prevent publication, but the authors should acknowledge this shortcoming in the text of the article.

The use of DropViz is mentioned. There should probably be a reference or web address.

I don't understand why the raw files have not been uploaded to the PRIDE repository. This should have been done already.

The text makes several statements about contaminant lists that were created from various experiments and subtracted. But I could not find one single clear statement that described clearly how a protein was designated as a contaminant. This should be made clear, particularly because there is a long list that has been designated as contaminants and they should not be made so without good reason.

I mentioned previously that gene names should not be used when referring to a protein, but there are still many instances when the names have not been used properly.

For example, "Actn4 is a filamentous actin-binding protein" is incorrect because Actn4 is a gene not a protein.

"Alpha-actinin-4 is a filamentous actin-binding protein" or "Actn4 encodes a filamentous actin-binding protein" is correct.

Reviewer #3 (Remarks to the Author):

The authors have responded well to all the reviewer's comments, with the addition of 2 key new experiments (in Figures 4 and 5). Overall, the manuscript is much improved, and I only have a few relatively minor points to consider.

1. The figure legends generally could be more informative.
2. L 24 – "proteomes"
3. L 616 – DIO abbreviation definition – move earlier
4. Fig S6a - and Fig 4a, Fig S8b,c – no discussion/citation in text
5. L 221 – "Additional experiments....." – such experiments were performed. What is the answer?
6. Fig S7b – non-alignment of replicate number with lanes
7. Was any attempt made to isolate SPN terminals from GPe (or SNr)?

Overall summary of response to review

We thank the Reviewers for continued support and for taking the time to re-evaluate the manuscript. We have addressed all comments. Changes are highlighted in the text, including the line numbers.

Reviewer #1 (Remarks to the Author):

The authors have satisfactorily addressed all of my previous questions. I appreciate the two entirely new experimental figures that directly answer questions regarding generality of this approach across cell types and the dynamic modulation of the proteome in response to chemogenetic activation.

We thank the Reviewer for taking the time to re-evaluate the manuscript and continued support.

Reviewer #2 (Remarks to the Author):

Overall the manuscript is greatly improved by the addition of a cell type comparison and a dynamic change due to stimulation. The manuscript will now be of high interest to the community, although mainly for its methodological aspects and cell type comparison. The dynamic protein changes don't make any new discoveries. The work is done to a high standard with good methodology and statistics, including the additions. Some minor outstanding issues are below.

We thank the Reviewer for taking the time to re-evaluate the manuscript. We have addressed all remaining minor issues.

I mentioned previously that competition from endogenously biotinylated proteins could interfere with protein extraction, particularly when the streptavidin is not in excess. The authors did not test this. The issue could affect the ranking of proteins in their data. I don't think this should prevent publication, but the authors should acknowledge this shortcoming in the text of the article.

We have included additional statements in the discussion section. Lines: 368–372

“Multiple technical replicates and exclusion lists can be used to increase proteome coverage. Pooling across animals may be necessary for smaller brain regions or for very low abundance subcellular compartments. The amount of beads can also be systematically adjusted to improve yield, because competition from endogenously biotinylated proteins could affect the ranking of proteins in the data. This could be particularly valuable for samples where correlative information is difficult to obtain from e.g., transcriptomics datasets. Here, we did not vary bead concentration, because we were able to identify many known positives for a given APEX construct, cell class, and activity states using our statistical analysis approach.”

The use of DropViz is mentioned. There should probably be a reference or web address.

We added DropViz reference and website to the method section. Lines: 762

I don't understand why the raw files have not been uploaded to the PRIDE repository. This should have been done already.

Our raw files have now been uploaded to the PRIDE repository with the project accession PXD022335 (<https://www.ebi.ac.uk/pride/archive/projects/PXD022335>). This dataset is now publicly accessible.

The text makes several statements about contaminant lists that were created from various experiments and substracted. But I could not find one single clear statement that described clearly how a protein was designated as a contaminant. This should be made clear, particularly because there is a long list that has been designated as contaminants and they should not be made so without good reason.

We added additional details on the contaminant list and justifications in the method section. Lines: 765–769

“A complete list of contaminants can be found in the Supplementary Data 4 tab 2. This contaminant list includes (i) proteins that are non-specifically enriched in the comparison between Cre+ and Cre- samples in Fig. 5 (i.e., proteins that did not pass the moderated t-test cutoff: 5% FDR and $\log_2FC > 0$), (ii) mouse keratin, and (iii) astrocyte-enriched proteins. In addition to non-specific binding contaminants, we decided to remove some abundant astrocyte-enriched proteins because in this experiment the goal is to compare two neuronal subtype proteomes.”

I mentioned previously that gene names should not be used when referring to a protein, but there are still many instances when the names have not been used properly.

For example, "Actn4 is a filamentous actin-binding protein" is incorrect because Actn4 is a gene not a protein. "Alpha-actinin-4 is a filamentous actin-binding protein" or "Actn4 encodes a filamentous actin-binding protein" is correct.

We recognize the Reviewer point about clarifying gene vs gene product. The authors use gene names to refer to their protein products to improve the consistency and readability between the manuscript and all Supplementary Data. A note has been added in the main text to clarify this for the reader. Lines: 193–194

"Proteins that were not statistically significantly enriched in the positive direction were discarded as non-specific binders (Fig. 3b, Supplementary Fig. 6a). Gene names will be used to refer to gene products for ease of reading. Some examples included known endogenously biotinylated proteins (*Acaca*, *Pccb*, *Pcca*, *Pc*, *Mccc1*). "

Reviewer #3 (Remarks to the Author):

The authors have responded well to all the reviewer's comments, with the addition of 2 key new experiments (in Figures 4 and 5). Overall, the manuscript is much improved, and I only have a few relatively minor points to consider.

We thank the Reviewer for continued support and for identifying minor remaining points, addressed below.

1. The figure legends generally could be more informative.

Figure legends have been revised throughout to include the number of independent replications and add more details on plot definition.

2. L 24 – "proteomes"

We have corrected this typo, highlighted in the text.

3. L 616 – DIO abbreviation definition – move earlier

We included the DIO abbreviation in two sections, for improved readability—Plasmid construction and AAV preparation and Stereotactic injections. Lines: 448 and 466

4. Fig S6a - and Fig 4a, Fig S8b,c – no discussion/citation in text

We have added additional details in the main text for Fig. 4a, Fig. S6a, and Fig. S8b,c.

Lines: 246

"To map both the cytosolic and the nuclear proteome of SPNs, we express APEX.NES and H2B.APEX in striatal *Drd1^{Cre}* and *A2a^{Cre}* neurons via neonatal viral transduction to target dSPNs and iSPNs, respectively (Fig. 4a)."

Lines: 193

"Proteins that were not statistically significantly enriched in the positive direction were discarded as non-specific binders (Fig. 3b, Supplementary Fig. 6a)."

Lines: 265–266

"Hierarchical clustering heatmaps of the NES and H2B show similarity across biological replicates, and conditions (Supplementary Fig. 8b,c)."

5. L 221 – "Additional experiments....." – such experiments were performed. What is the answer?

We added further description to the result section below. Lines: 274–277

"Therefore, we consider proteins differentially expressed with p-value <0.05. *Drd1* and *A2a* receptors were correctly identified in the cytosolic proteome of dSPNs and iSPNs, respectively (highlighted in Fig. 4g). As for the *Drd1^{Cre+}* nuclear proteome detected in the prior dataset (Fig. 2–3), the majority of highly abundant nuclear proteins were general nuclear markers for both SPNs, measured by *A2a^{Cre}* – *Drd1^{Cre}* comparison (e.g., *Hnrnpa2b1*, log2FC = -0.043, p = 0.607, and *Hnrnpd*, log2FC = -0.011, p = 0.89)."

6. Fig S7b – non-alignment of replicate number with lanes

The lane numbering in Fig S7b has been corrected.

7. Was any attempt made to isolate SPN terminals from GPe (or SNr)?

We did not perform this experiment, but agree that this is an important future direction, now noted explicitly in the discussion section. Lines: 425–429

“While the current work demonstrates activity-dependent cell-type specific profiling for the nuclear proteome, the method is modular and applicable to other cellular compartments. Although we did not isolate SPN terminals from GPe or SNr in this study, future work investigating the proteomes of SPN axons may uncover differences that relate to segregated projection targeting, developmental dynamics, or changes in disease models.”